# Shaping active matter from crystalline solids to active turbulence

Qianhong Yang [1,4], Maoqiang Jiang[2,4], Francesco Picano [3] & Lailai Zhu [1] ✉

Active matter drives its constituent agents to move autonomously by harnessing free energy, leading to diverse emergent states with relevance to both biological processes and inanimate functionalities. Achieving maximum reconfigurability of active materials with minimal control remains a desirable yet challenging goal. Here, we employ large-scale, agent-resolved simulations to demonstrate that modulating the activity of a wet phoretic medium alone can govern its solid-liquid-gas phase transitions and, subsequently, laminar-turbulent transitions in fluid phases, thereby shaping its emergent pattern. These two progressively emerging transitions, hitherto unreported, bring us closer to perceiving the parallels between active matter and traditional matter. Our work reproduces and reconciles seemingly conflicting experimental observations on chemically active systems, presenting a unified landscape of phoretic collective dynamics. These findings enhance the understanding of long-range, many-body interactions among phoretic agents, offer new insights into their non-equilibrium collective behaviors, and provide potential guidelines for designing reconfigurable materials.

Active matter represents a class of material systems comprised of autonomous units capable of converting free energy into mechanical work[1,2]. The collective motion of these units, mediated by their interactions, can bring in fascinating self-organization, pattern formation, and coherent activities[3–5]. Their emergence and maintenance are crucial in biological systems[6,7], inspiring the development of synthetic, autonomous agents as the foundation for reconfigurable, functional materials[8,9]. A key objective of designing such bio-inspired systems is to maximize their reconfigurability with minimal control. Prior research has demonstrated controlling activity to achieve either phase change in active matter[10,11] or laminar-turbulent transition in active fluids[12,13], but not both. Here, we find that remarkably, tuning the activity of a phoretic medium alone can control its solid–liquid–gas phase transitions and, subsequently, laminar-turbulent transitions in fluid phases. Through large-scale, agent-resolved simulations, we investigate suspensions of isotropic phoretic agents (IPAs) epitomized by active droplets[14–27] and camphor surfers[28], explicitly resolving their many-body hydrochemical interactions. Our dual consideration of

long-range hydrodynamic and chemical interactions enables not only reproducing characteristic collective behaviors of IPAs observed in the lab, but also reconciling seemingly divergent experimental observations—active crystallization or turbulence. The unified landscape of phoretic collective dynamics is unattainable by resolving either the hydrodynamic or chemical interaction alone.

We focus on a two-dimensional (2D) paradigmatic system of IPAs. An IPA, unlike Janus colloids[29,30], acquires autonomous propulsion through instability, but otherwise in a stable stationary state. For instance, an active droplet undergoing uniform surface reaction exchanges solutes with the ambient, causing an isotropic solute distribution (Fig. 1b). A perturbation inducing a Marangoni interfacial flow advects the solute, amplifying itself. When the ratio of the destabilizing advection to the stabilizing solute diffusion—characterized by Péclet (Pe) number, exceeds a threshold, instability emerges, driving the droplet to swim steadily (Fig. 1c, left); increasing Pe may trigger its chaotic movement[31–36] (Fig. 1c, right). Similarly, this mechanism allows camphor disks to swim continuously or intermittently[37,38].

[1]Department of Mechanical Engineering, National University of Singapore, Singapore, Singapore. [2]School of Naval Architecture, Ocean and Energy Power Engineering, Wuhan University of Technology, Wuhan, Hubei, PR China. [3]Department of Industrial Engineering and CISAS "G. Colombo", University of Padova, Padova, Italy. [4]These authors contributed equally: Qianhong Yang, Maoqiang Jiang. ✉e-mail: lailai_zhu@nus.edu.sg

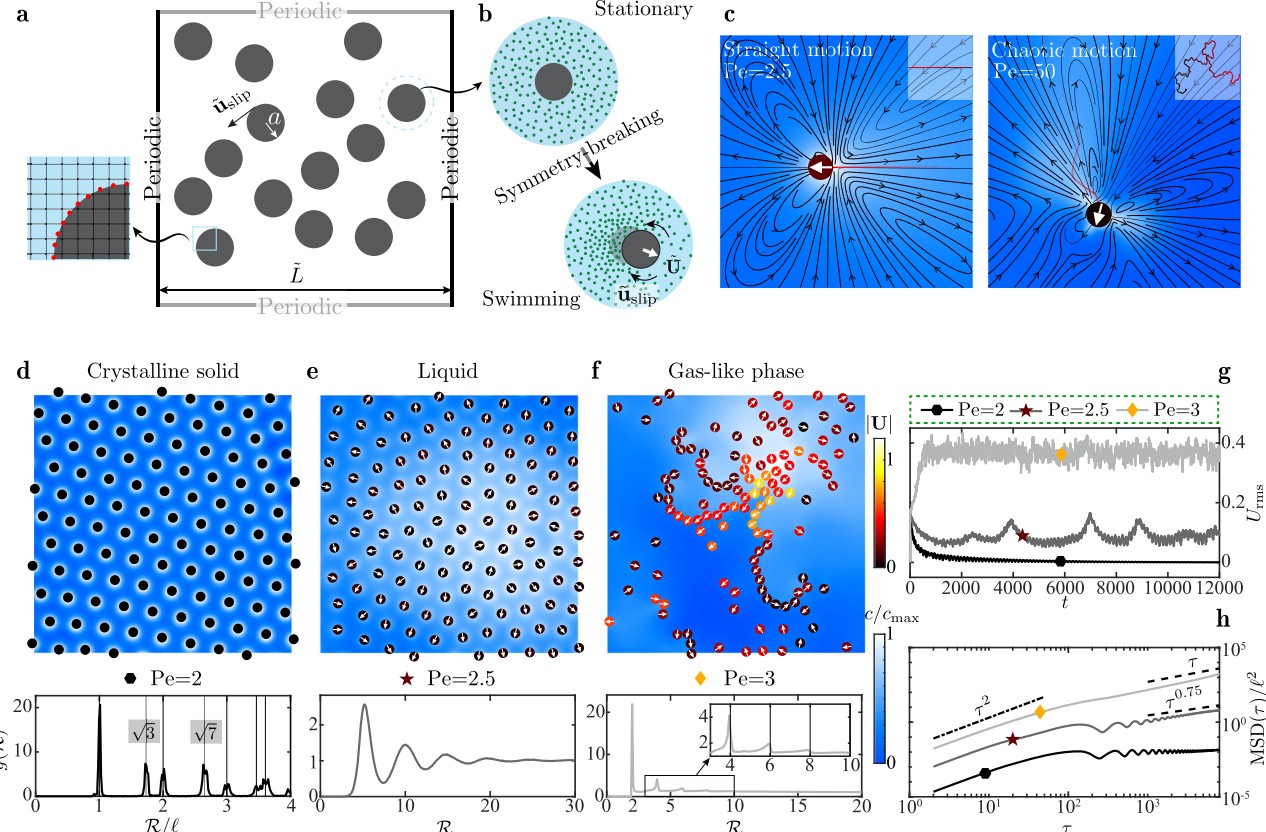

**Fig. 1 | Activity-induced phase transition. a** $N$ circular phoretic disks of radius $a$ freely swimming in a periodic square domain of size $\tilde{L}$. The notation $\tilde{}$ represents dimensional variables throughout. The side panel illustrates the numerical discretization (Supplementary information Sec. II). **b** sketch of a single phoretic disk attaining a swimming velocity, $\tilde{\mathbf{U}}$, spontaneously via instability. Green dots indicate a chemical solute emitted by the disk, while $\tilde{\mathbf{u}}_{slip}$ denotes its surface slip velocity induced by the chemical gradient. **c** A swimming disk of low (left) or high (right) activity—implied by the Péclet number Pe—follows a straight or chaotic trajectory, respectively. The colormap shows the scaled solute concentration $c/c_{max}$ and the insets depict the trajectories. The streamlines surrounding the swimmer at Pe = 2.5

demonstrates its pusher-like dipolar signature, as previously identified numerically[31] and experimentally[88]. **d–f** Disks with an area fraction $\phi = 0.12$ self-organize into hexagonal solid (Pe = 2), liquid (Pe = 2.5), and gas-like (Pe = 3) phases depicted in a quarter of the domain. Disks are colored by their swimming speed $|\mathbf{U}|$ and the arrow indicates the instantaneous direction of $\mathbf{U}$. The second row displays the corresponding pair correlation function $g(\mathcal{R})$. Here, $\mathcal{R}$ denotes the inter-disk distance and $\ell = \left(2\pi\phi^{-1}/\sqrt{3}\right)^{1/2}$ represents the lattice constant. **g, h** Root-mean-square (RMS) disk velocity $U_{rms}$ versus time and the scaled mean square displacement (MSD) versus the time lag $\tau$, respectively, for varying Pe. See Source data.

We consider $N$ overdamped disks of radius $a$ in a doubly periodic square domain of size $\tilde{L}$ (Fig. 1a). Hereinafter, $\tilde{}$ denotes dimensional variables. The disks uniformly emit a chemical solute of molecular diffusivity $\mathcal{D}$ at a constant rate $\mathcal{A} > 0$. The solute distribution $\tilde{c}\left[\tilde{\mathbf{r}} = (\tilde{x}, \tilde{y}), \tilde{t}\right]$ causes a slip velocity $\tilde{\mathbf{u}}_{slip} = \mathcal{M}\left(\mathbf{I} - \mathbf{nn}\right) \cdot \tilde{\nabla}\tilde{c}$, with $\mathcal{M}$ the mobility coefficient and $\mathbf{n}$ the outward normal at the disk surface. We vary the phoretic activity Pe $= \mathcal{A}\mathcal{M}a/\mathcal{D}^2$ and area fraction $\phi = \pi N/L^2$ ($L = \tilde{L}/a$) of disks to explore their collective dynamics, solving dimensionless physicochemical hydrodynamics involving the fluid velocity $\mathbf{u}$ and pressure $p$, and solute concentration $c$, see "Methods" and Supplementary information (SI) Sec. II.

## Results

### Crystalline solid, liquid, and gas-like phases

A disk suspension of $\phi = 0.12$ exhibits Pe-dependent collective patterns (Fig. 1d–f). At Pe = 2, disks undergo transient, disordered motion that decays in time. Hence, their root-mean-square (RMS) velocity $U_{rms} = \sqrt{N^{-1}\sum_{k=1}^{N}\mathbf{U}_k^2}$ ($\mathbf{U}_k$ is the translational velocity of the $k$-th disk) eventually approaches zero, when they self-organize into a stationary hexagonal lattice resembling a 2D crystalline solid (Fig. 1d and Supplementary Video 1). This resemblance is supported by the pair correlation function $g(\mathcal{R})$ versus the inter-disk distance $\mathcal{R}$ (SI Sec. I) depicted in Fig. 1d, which reveals the signature of a 2D hexagonal

crystal with a lattice constant $\ell = (2\pi\phi^{-1}/\sqrt{3})^{1/2}$: $g(\mathcal{R})$ peaks around discrete inter-disk distances $\mathcal{R}/\ell = 1, \sqrt{3}, 2, \sqrt{7}, \ldots$. Importantly, the lattice constant $\ell \approx 5.5$ considerably larger than the disk diameter implies that this solid shares similarity with the Wigner crystal constituting electrons, as theoretically predicted[39,40] and directly visualized in experiments very recently[41]. Unlike the long-range Coulomb force causing the electronic Wigner crystallization, the chemorepulsion among phoretic disks[18,25,42] creates the active Wigner crystal. This phenomenon agrees with the experimental observations on camphor surfers[43] and active droplets[19], as well as numerical predictions for the former[44]. Such active Wigner crystals are distinct from the hexagonal closed-packed crystallization commonly reported in other active suspensions[45–52], where active units tend to achieve the highest packing fraction akin to the atomic arrangement in graphene layers and some metals.

Increasing the activity to Pe = 3, the disks freely swim like Brownian gas molecules (Fig. 1f and Supplementary Video 2), with randomly fluctuating speed $U_{rms}$ (Fig. 1g). After a transient period, this fluctuating motion exhibits normal diffusion, as evidenced by the mean square displacement (MSD) of disks (Fig. 1h). Dynamic arch-shaped chains of closely spaced disks also appear, similar to chains of active droplets observed experimentally[14,19]. These chains form, collide, annihilate, and reform continuously (Supplementary Video 2). This dynamic

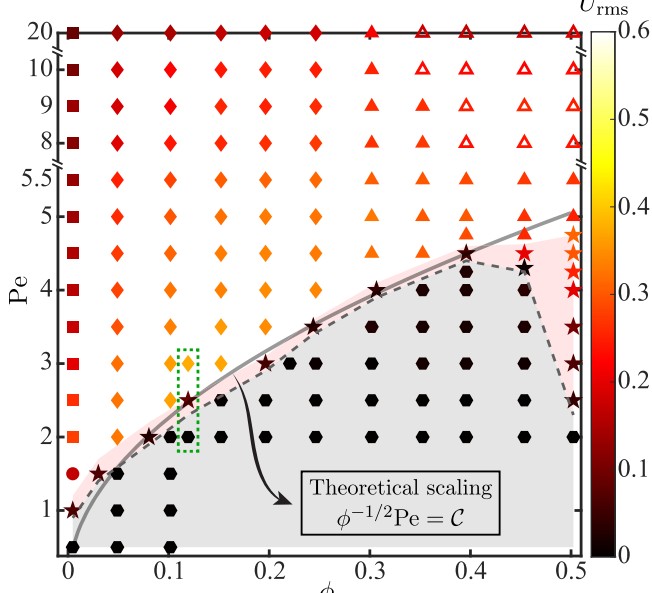

**Fig. 2 | Phase diagram over the area fraction $\phi$ and activity Pe of phoretic disks.** The disks form crystalline solids (hexagon), liquids (star), or gas-like phases consisting of four classes: no pattern (circle); single stable chain (square); dynamic chaining (diamond); and self-organized coherent flows (triangle). Symbols are colored by the time-averaged RMS velocity $\overline{U_{\text{rms}}}$ of disks. The solid curve represents the theoretical prediction Eq. (2) of the solid-liquid transition, where $\mathcal{C}$ denotes a fitting parameter. The dashed box encloses the three phases showcased in Fig. 1.

chaining differs from the single stable chains observed at lower fraction $\phi$, which originate from a balance between inter-disk chemorepulsion and hydrodynamic attraction (SI Sec. III). Overall, this suspension reveals features of both a gas phase and chain formation, as statistically evinced in Fig. 1f: $g(\mathcal{R}) \approx 1$ in most region of $\mathcal{R} \geq 2$, characteristic of a homogeneous molecular distribution; it develops peaks with decreasing magnitudes approximately at even multiples of the disk radius, i.e., $\mathcal{R} \approx 2, 4$, and $6$—that signify chain forming.

Between the solid and gas-like phases, a liquid phase appears at Pe = 2.5 (Fig. 1e and Supplementary Video 3). The disks vibrate, move around, and slide over each other mimicking the behavior of liquid molecules. The $U_{\text{rms}}$ displays a super-oscillatory time evolution (Fig. 1g), and the MSD $\propto \tau^{0.75}$ exhibits subdiffusive dynamics (Fig. 1h). This anomalous diffusion, unlike the normal diffusion of classical liquids, is typical of crowded fluids containing suspended macro-molecules or colloids that act as obstacles[53,54]. The identified liquid phase is confirmed by the pair correlation function $g(\mathcal{R})$ depicted in Fig. 1e, where several oscillations at short distances $\mathcal{R}$ attenuate with increasing $\mathcal{R}$, indicating the diminished long-range order characteristic of liquids.

**Phase diagram and theoretical prediction of the solid–liquid transition**
Displaying the time-averaged mean disk velocity $\overline{U_{\text{rms}}}$ versus the fraction $\phi$ and activity Pe in Fig. 2, we reveal the $\phi$ − Pe phase diagram of the suspension. Besides the solid-liquid-gas transitions, we divide the gas-like phase into four regimes: no pattern; single stable chain; dynamic chaining (Fig. 1f); and self-organized large-scale flows exhibiting instability, transition and active turbulence, which will be discussed later; the first two regimes are analyzed in SI Sec. III. For now, we focus on theoretically explaining the solid-liquid transition.

Upon increasing Pe to cross a threshold (dashed curve in Fig. 2), the suspension transitions from a quiescent hexagonal pattern to an unsteady motion. This change is in reminiscent of the stationary-to-swimming instability of an IPA with sufficient activity Pe. The

theoretically predicted onset of instability for a single IPA[32,55–57] provides inspiration for a prediction that characterizes the collective instability of our suspension.

We consider the hexagonal tiling of disks as the unperturbed state. Following Pe $= \mathcal{A}\mathcal{M}a/\mathcal{D}^2$, which determines the instability of a single disk, we define $\text{Pe}_{\text{col}}$ for the collective instability of many featuring $\phi$. From the dimensional advection–diffusion equation ("Methods"), we infer

$$\text{Pe}_{\text{col}} = \frac{[\tilde{\mathbf{u}}]\left[\tilde{\boldsymbol{\nabla}}\tilde{c}\right]}{\mathcal{D}\left[\tilde{\boldsymbol{\nabla}}^2\tilde{c}\right]}, \tag{1}$$

using $[\cdot]$ to represent the characteristic scales. As for a single disk, $[\tilde{\mathbf{u}}] = \mathcal{A}\mathcal{M}/\mathcal{D}$ remains. Further limited to the low-$\phi$ regime corresponding to a large inter-disk distance $\tilde{\ell} \gg a$ ($\ell \gg 1$), we regard the disks as points and thus approximate $|\tilde{\boldsymbol{\nabla}}\tilde{c}|$ at the disk center by its exact value $\mathcal{A}/\mathcal{D}$ at the disk boundary. At the midpoint of two neighboring disks, $\tilde{\boldsymbol{\nabla}}\tilde{c} = \mathbf{0}$ due to the hexagonal symmetry (SI Sec. III.A). Hence, $|\tilde{\boldsymbol{\nabla}}\tilde{c}|$ decays from $\mathcal{A}/\mathcal{D}$ to zero within the distance $\tilde{\ell}/2$, leading to $\left[\tilde{\boldsymbol{\nabla}}^2\tilde{c}\right] = 2\mathcal{A}/(\mathcal{D}\tilde{\ell})$ and subsequently

$$\text{Pe}_{\text{col}} = \left(\frac{\pi}{2\sqrt{3}}\right)^{1/2} \phi^{-1/2}\, \text{Pe} \propto \phi^{-1/2}\text{Pe}. \tag{2}$$

Assuming that the instability occurs when $\phi^{-1/2}$Pe exceeds a constant $\mathcal{C}$, we obtain, via fitting, the predicted threshold $\phi^{-1/2}$ Pe $= \mathcal{C}$ (solid curve in Fig. 2) that matches the actual one reasonably when $\phi \lessapprox 0.4$.

**Two-dimensional melting via a hexatic phase**
Further tuning Pe as the activity-induced effective temperature, we scrutinize the solid-liquid transition at $\phi = 0.12$ as a melting scenario. We identify an intermediate phase that is neither solid nor liquid. As indicated in Fig. 3a, the structure factor $S(\mathbf{q})$ (SI Sec. I.C) for Pe = 2.3 exhibits definite Bragg peaks with six-fold symmetry, revealing the formation of a crystalline solid. Raising the effective temperature to Pe = 2.5, the Bragg peaks are almost smeared out, leaving an isotropic ring pattern of $S(\mathbf{q})$ with insignificant orientational symmetry. This suggests that the suspension has melted, reaching a liquid state. At Pe = 2.4, the translational order is lost while six-fold orientational symmetry preserves. This intermediate state corresponds to a hexatic phase between the solid and liquid, as described by the celebrated Kosterlitz, Thouless, Halperin, Nelson, and Young (KTHNY) theory[58–60]. This theory, originally built for equilibrium systems, has also been tested upon non-equilibrium counterparts of active agents[50,61–64].

Within the KTHNY theory, melting of 2D crystals is mediated by progressive creation of topological defects, as illustrated in Fig. 3b for our case. When Pe = 2.3, we detect bound dislocation pairs. Increasing the activity, we observe the dissociation of such pairs into free dislocations at Pe = 2.4 (Supplementary Video 4), and the unbinding of dislocations into isolated disclinations at Pe = 2.5 (Supplementary Video 5). The two successive processes drive the solid-to-hexatic and hexatic-to-liquid transitions, respectively, which highlight the signature two-step storyline of the KTHNY framework.

To confirm this scenario quantitatively, we examine, close to the phase transition, the correlation functions $g_{\mathbf{q}_0}(\mathcal{R})$ and $g_6(\mathcal{R})$ of the translational and orientational order (SI Sec. I.C). As illustrated in Fig. 3c, the former $g_{\mathbf{q}_0}(\mathcal{R}) \propto \mathcal{R}^{-\eta}$ with $\eta \approx 1/25$ when Pe = 2.3 (SI Sec. III.E). This algebraic scaling, suggestive of the quasi long-range translational order, typifies a 2D solid, with the power-law exponent $-\eta \geq -1/3$[65]. Conversely, $g_{\mathbf{q}_0}(\mathcal{R})$ decays exponentially at Pe = 2.35, visually faster than the limiting behavior $g_{\mathbf{q}_0} \propto \mathcal{R}^{-1/3}$ of solids[65]. The corresponding short-range translational order implies a liquid/hexatic phase when Pe $\geq$ 2.35. Akin to $g_{\mathbf{q}_0}(\mathcal{R})$, we depict in Fig. 3d the orientational order correlation function $g_6(\mathcal{R})$ that features three scaling

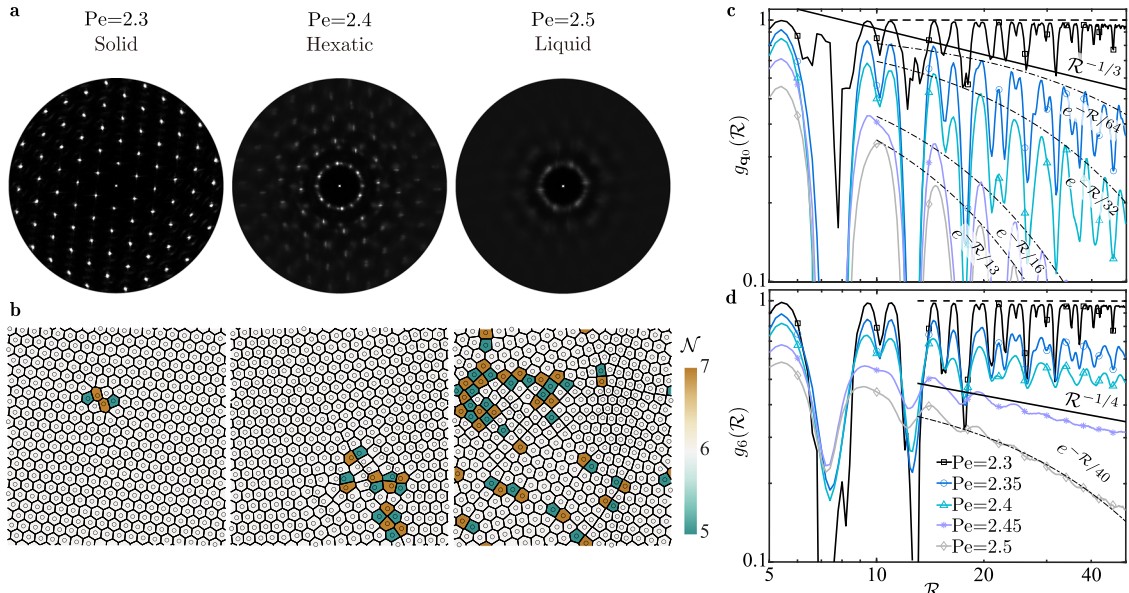

**Fig. 3 | Defect-mediated melting via a hexatic phase. a** Structure factor $S(\mathbf{q})$ of a $\phi = 0.12$ disk suspension at three Pe values, indicative of a growing effective temperature. **b** Evolving topological defects associated with the three phases in **a** showing a bound dislocation pair, two free dislocations, and coexisting dislocations and disclinations, consecutively. Disks with $\mathcal{N} = 5$ and 7 neighbours are marked green and yellow, respectively. **c** Translational order correlation function $g_{\mathbf{q}_0}(\mathcal{R})$ at Pe $\in [2.3, 2.5]$; the horizontal dashed line indicates a constant, non-decaying $g_{\mathbf{q}_0}(\mathcal{R})$. **d** Similar to **c** but for the orientational order correlation function $g_6(\mathcal{R})$.

laws: at the lowest temperature Pe = 2.3, the peaks of $g_6(\mathcal{R})$ remain approximately constant, evidencing the long-range orientational order of 2D solids; in the other limit Pe = 2.5, $g_6 \propto \exp(-\mathcal{R}/40)$ indicates the short-range orientational order possessed by liquids; at intermediate temperatures, $g_6 \propto \mathcal{R}^{-\eta'}$, where $-\eta' > -1/4$ at Pe = 2.35 and 2.4 but $-\eta' \lessapprox -1/4$ when Pe = 2.45. Recalling that the orientational order of a hexatic phase decays algebraically with the exponent $-\eta' \geq -1/4$ according to the KTHNY prediction[59,66], we confirm the presence of a hexatic phase at Pe $\in [2.35, 2.4]$. Although this specific regime may vary marginally upon doubling the domain size $L$, the overarching two-step melting process remains consistent (SI Sec. III.E).

**Instability, transition, and active turbulence**

Viewing the melted disk suspension of $\phi = 0.5$ as a continuum active fluid, we examine the spatial-temporal evolution of its Eulerian velocity $\mathbf{U}_E = (U_E, V_E)$ and area fraction $\phi_E$ (SI Sec. I.D). Here, $\Phi_E = (\phi_E - \phi)/\phi$ effectively measures the "pressure" of the active fluid.

At Pe = 3, the domain-averaged ($\langle \rangle_{xy}$) speed $\langle |\mathbf{U}_E| \rangle_{xy}$ and absolute pressure $\langle |\Phi_E| \rangle_{xy}$ approach nearly zero after a transient decline (Fig. 4b), reaching a quiescent flow state (laminar). Conversely, such Eulerian quantities at Pe = 5 grow and level off at considerable values, indicating a self-organized active flow of disks (Fig. 4a and Supplementary Video 6). The flow appears as large-scale waves indicated by the time-periodic evolution of the $x$-averaged pressure $\langle \Phi_E \rangle_x$ and velocity $\langle V \rangle_x$ (Fig. 4c), and the undulating kymograph of $\Phi_E|_{x=\frac{L}{2}}$ at the middle line $x = \frac{L}{2}$ (Fig. 4d, top). The time evolution of wave (Fig. 4e, top) shares a qualitative similarity with the occurrence of an oscillatory hydrodynamic instability via a Hopf bifurcation—an infinitesimal disturbance growing exponentially saturates to a periodic state. The similarity hints at the resemblance between this activity-induced quiescence-oscillation transition and inertia-triggered hydrodynamic instabilities, as recognized for polar active fluids[13]; that instability occurs when Reynolds (Re) number inversely related to the kinematic viscosity grows. Analogously, increasing Pe here enables the transition from a more viscous to a less viscous phase (solid-to-liquid-to-gas) as shown in Fig. 2, indeed enlarging the effective Re of the active fluid and thus promoting its instability.

As Pe is raised from 5 to 10, the undulatory dynamics becomes irregular (Fig. 4e, bottom). In parallel, the kymograph of $\Phi_E|_{x=\frac{L}{2}}$ exhibits a disrupted wave pattern (Fig. 4d, bottom), seemingly indicative of a secondary instability that is known to trigger the breakdown of streaks and waves in classical hydrodynamics[67]. Moreover, the wave breakdown here is reflected by the emergent clusters of disks showcased in Fig. 4f (Supplementary Video 7), which imply fluctuations of the effective pressure $\Phi_E$. The cluster size distribution $\mathcal{P}(N_{clu})$ shown in Fig. 4g further evidences the clustering events at Pe = 10 quantitatively.

At a even higher activity Pe = 20, oscillatory patterns fade significantly in favor of emerging vortical structures (Fig. 5a and Supplementary Video 8), hinting the occurrence of active turbulence. Concurrently, enhanced clustering is evident, potentially resulting from the turbulence, as identified in polar active fluids[68].

Despite their Lagrangian appearance (SI Sec. III.F), Fig. 5b manifests the vortical structures by the Eulerian flow streamlines and vorticity field $\boldsymbol{\omega}_E \cdot \mathbf{e}_z$. This Eulerian description enables using the Okubo-Weiss parameter (SI Sec. I.D) to quantify the vortex sizes, which are shown in Fig. 5d to feature a Gaussian probability density function (PDF). Besides, we pinpoint the largest cluster of disks and identify its effective size $l_{clu}(t)$ as the longest disk-disk distance (Fig. 5c). The evolution of $l_{clu}$ depicted in Fig. 5e reveals its time-averaged value $\ell_{clu} \approx 87$.

Apart from the graphic depiction, we compare the statistics of the turbulent disk motion at Pe = 20 to those at Pe = 5. As shown in Fig. 5g, the PDF of the velocity component $U$ is approximately Gaussian in both cases. Compared to the less active scenario (Pe = 5), PDF at Pe = 20 departs slightly more from the Gaussian distribution attributed to the corresponding packed and dissipative clusters[69]. Akin to understanding inertial turbulence, we examine the velocity difference $\Delta \mathbf{U}(t, \mathbf{R}, \mathcal{R}) = \mathbf{U}(t, \mathbf{R} + \mathcal{R}) - \mathbf{U}(t, \mathbf{R})$ between two disks separated by a displacement $\mathcal{R}$, focusing on statistics of its longitudinal component $\Delta U^{\|} = \Delta \mathbf{U} \cdot \mathcal{R}/|\mathcal{R}|$. Reducing the separation $\mathcal{R}$ from 20 to 5, Fig. 5h, i suggest that the PDF of $\Delta U^{\|}$ deviates more from the Gaussian profile. The deviation is more pronounced at Pe = 20, as reflected by the fat-tailed distribution for $\mathcal{R} = 5$ (Fig. 5i). The fat tails manifest a stronger probability of high-amplitude extreme events associated with small-scale intermittent processes as well recognized in canonical

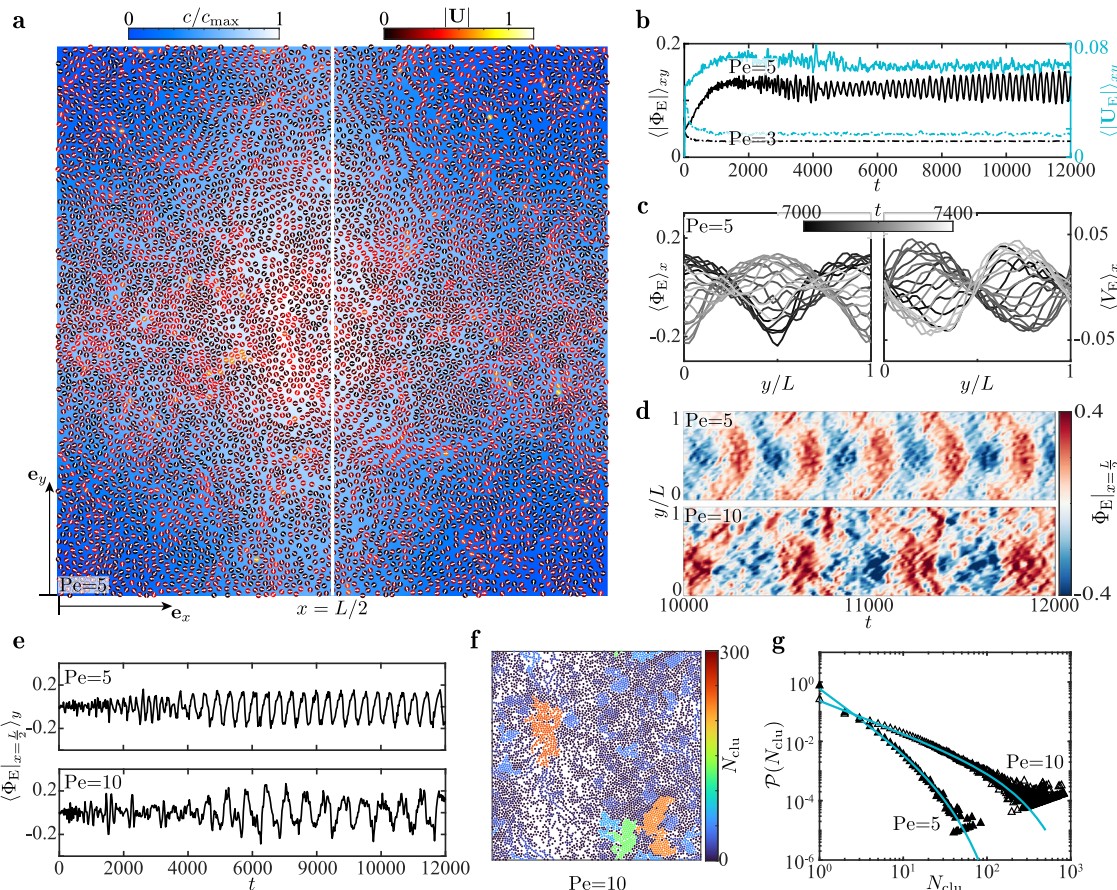

**Fig. 4 | Instability and transition of the active fluid with $\phi = 0.5$. a** An oscillatory active flow self-organizes via instability at Pe = 5, when the dense population of disks shifts periodically between the corners (this snapshot) and the center (SI Sec. III.F) of the domain. **b** History of domain-averaged absolute pressure $\langle|\Phi_E|\rangle_{xy}$ and speed $\langle|\mathbf{U}_E|\rangle_{xy}$ of the active fluid implies the absence and emergence of a self-organized flow at Pe = 3 and Pe = 5, respectively. **c** Wave-like spatiotemporal evolution of $x$-averaged pressure $\langle\Phi_E\rangle_x$ and velocity component $\langle V_E\rangle_x$ at Pe = 5.

**d** Corrugated kymograph of $\Phi_E|_{x=\frac{L}{2}}$ sampled along the median $x = L/2$ when Pe = 5 (upper) versus its disrupted counterpart for Pe = 10 (lower). **e** History of $\langle\Phi_E|_{x=\frac{L}{2}}\rangle_y$ depicts the evolution and saturation of disturbances mimicking those of their canonical hydrodynamic analog. **f**, when Pe = 10, disks form clusters that break-down the wave patterns. The clusters are characterized by the number $N_{clu}$ of their constituting disks. **g** Cluster size distribution function $\mathcal{P}(N_{clu})$, and its fitted curves following $\mathcal{P}(N_{clu}) = \mathcal{C}_1 N_{clu}^{-\mathcal{C}_2} \exp\left(-N_{clu}/N_{clu}^{\dagger}\right)$ defined in SI Sec. I.D. See Source data.

turbulence[70,71] and also highlighted very recently in a dry active system[72]. Here, they are linked to the formation of intense vortical structures yielding a larger relative velocity of two disks.

Focusing on the active turbulence at Pe = 20, we display in Fig. 5j the spectrum $E(\hat{q})$ of its kinetic energy $\langle\mathbf{U}^2\rangle/2$, versus the modified wavenumber (SI Sec. I.E), $\hat{q} = 2\pi/(\mathcal{R} - 2)$. The spectrum scales as $E(\hat{q}) \sim \hat{q}^{-5/3}$ and $E(\hat{q}) \sim \hat{q}^{-2/3}$, respectively, at small (large $\hat{q}$) and intermediate length scales. Furthermore, we observe a regime with a positive scaling exponent at large length scales (small $\hat{q}$). Specifically, the scaling $E(\hat{q}) \sim \hat{q}^{4/3}$ is pronounced for an enlarged domain size of $L = 400$. This regime connects with the $E(\hat{q}) \sim \hat{q}^{-2/3}$ regime via a peak that signifies the maximal energy injection[73,74]. Notably, the whole spectrum spanning these three regimes mirrors, in its shape, that of the experimentally observed active nematic turbulence[75]. In the latter case, the $E(q) \sim q^{-1}$ regime is confined to an intermediate length scale, above a vortex size $\ell_{vtx}$, and below a viscous length scale where $E(q)$ peaked. Analogously, our intermediate $E(\hat{q}) \sim \hat{q}^{-2/3}$ regime is bounded likewise: the lower bound exceeds the mean vortex size $\ell_{vtx} \approx 8$ identified in Fig. 5d; the upper bound, where the energy spectrum also approximately peaks, aligns closely to the time-averaged size $\ell_{clu} \approx 87$ of the largest cluster (Fig. 5e).

We note that our simulations have incorporated a weak yet finite fluid inertia of Re = 0.5 to approximate Stokes flow (Methods). However, inertia is not the primary factor driving our active turbulence, contrary to other configurations[76–78] featuring a

dominant inertial effect. Specifically, ref. 76 demonstrates that a suspension of rotating disks exhibits consistent collective behaviors at varying Re up to $\approx 0.6$, transitioning to chaos at Re $\gtrsim 5$ (see their Figure 7c).

On the other hand, ref. 78 highlights the high sensitivity of the emerging active turbulence to Re even below 0.1, unlike the weak Re-dependence depicted here (SI Sec. III.F). To rationalize the discrepancy, we first emphasize that both settings involve a driving nonlinearity and an auxiliary counterpart. In that study, an electric field of magnitude $\tilde{E}$ above a threshold $\tilde{E}_c$ drives an electro-hydrodynamic instability quantified by $\gamma = \tilde{E}/\tilde{E}_c$. In our setting, the driving nonlinearity arising from the phoretic transport causes a hydrochemical instability characterized by Pe. Despite their different driving mechanisms, both studies feature the same auxiliary nonlinearity: inertia. The reason why the strong Re-dependency in ref. 78 is absent here stems from a distinction in the strength of the driving nonlinearity relative to the inertial one. When the driving nonlinearity is comparable or even weaker than its inertial counterpart, the driver could feasibly intensify the impact of changing Re, whereas a dominant driving nonlinearity may mitigate this impact. Reference[78] adopts a nonlinearity level of $\gamma = 1.1$, just above the instability threshold: $\gamma = 1$, representing a weak driving nonlinearity. Conversely, we demonstrate active turbulence at Pe = 20, a value far above the threshold, Pe $\approx 0.5$. Hence, our driving nonlinearity substantially exceeds the inertial counterpart, implying that variations in Re around unity may wield

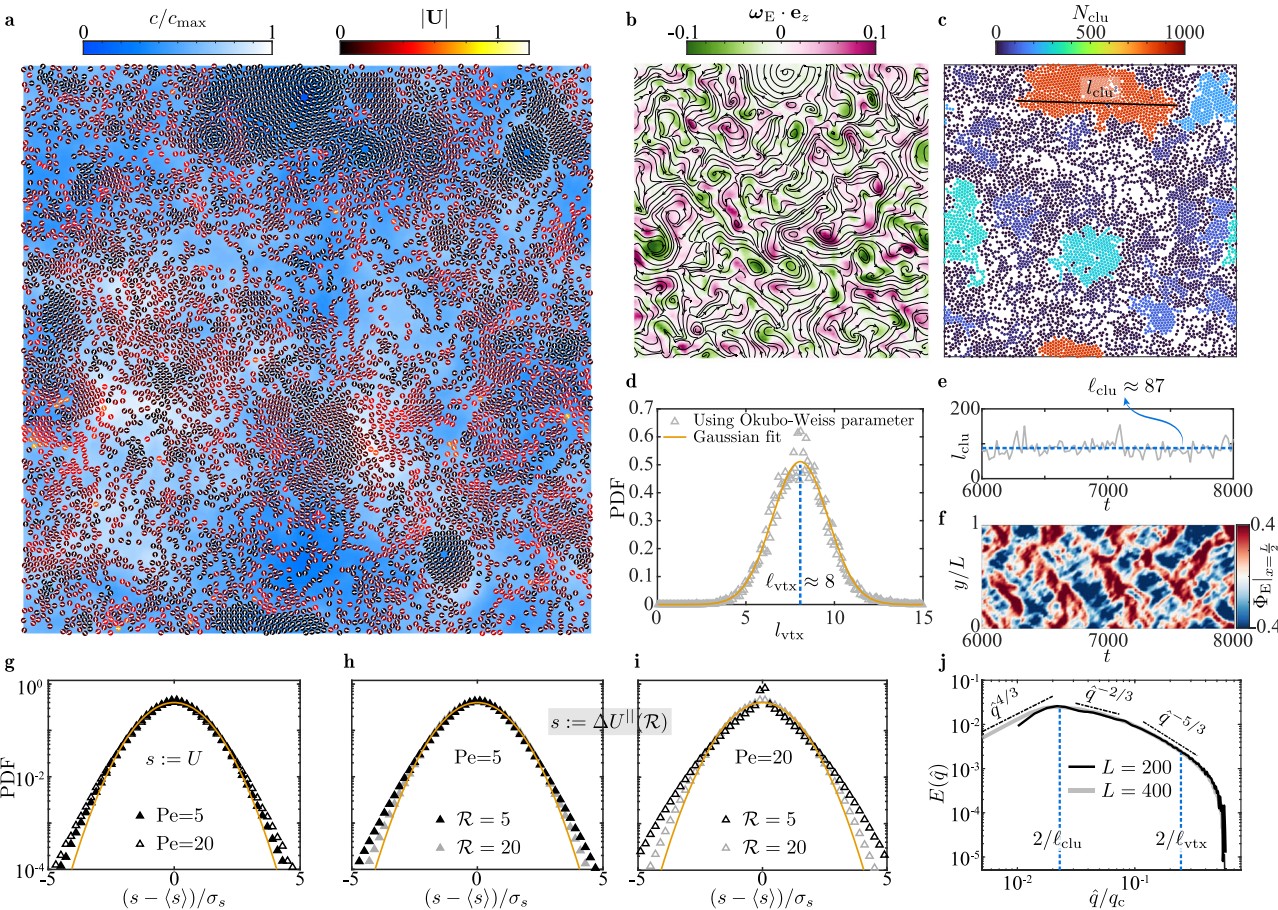

**Fig. 5 | Active turbulence. a** Instantaneous motion of disks colored by their speed |**U**| and concentration field $c/c_{max}$. Here, Pe = 20, $\phi$ = 0.5, and the domain size is $L$ = 200. **b**, vortex-showing streamlines and vorticity component $\boldsymbol{\omega}_E \cdot \mathbf{e}_z$ of the continuum flow field $\mathbf{U}_E$. **c** Clusters characterized by the number $N_{clu}$ of their constituting disks and the size $l_{clu}$ of the largest cluster. **b, c** showing the full domain are scaled versus **a**. **d**, probability density function (PDF) of the vortex size $l_{vtx}$ with a mean of $\ell_{vtx} \approx 8$. **e** History of $l_{clu}$ and its time-averaged value $\ell_{clu} \approx 87$. **f**, kymograph of $\Phi_E|_{x=\frac{L}{2}}$ implies chaos and fading oscillatory patterns. **g** PDF of the disk' velocity component $U$. **h**, PDF of the longitudinal velocity difference $\Delta U^{\parallel}(\mathcal{R})$ between two

disks separated by a distance $\mathcal{R}$ when Pe = 5. **i** Similar to (**h**) but for Pe = 20. In **g–i**, $\langle s \rangle$ and $\sigma_s$ denote the average and standard deviation of a random variable $s$, respectively; the curve represents the unit-variance Gaussian function $1/\sqrt{2\pi} \exp(-s^2/2)$. **j** Kinetic energy spectrum $E(\hat{q})$ versus the modified wavenumber $\hat{q}$ (SI Sec. I.E), with an additional dataset incorporated for an expanded domain where $L$ = 400. Here, $q_c = \pi$ is the characteristic wavenumber corresponding to the disk diameter. The left vertical line indicates the size, $\ell_{clu} \approx 87$ of the largest cluster, while the right line marks the mean vortex size, $\ell_{vtx} \approx 8$; both pertain to the case of $L$ = 200. See Source data.

minimal influence, as indeed supported by additional simulations (SI Sec. III.F).

## Discussion

By regarding its activity Pe as the analog of temperature, we showcase a wet active matter preserving all thermodynamic phases, as previously depicted for dry active matter[50,61]. Moving ahead by increasing this activity analogous to Reynolds number, the fluid phase exhibits progressively, a quiescent laminar state, waves via an oscillatory instability, clusters that break the waves down signifying transition, and finally, vortical structures suggestive of active turbulence. This progression highlights a stronger phenomenological resemblance between active and classical fluids in their laminar-turbulent transition than previously recognized.

Controlling both the phase transition and laminar-turbulent transition by activity solely is remarkable, yet not an isolated observation. Prior experiments on camphor swimmers have independently demonstrated their crystalline solid state[43] and turbulent-like collective activity[79], indeed indirectly supporting our unified interpretation of these dual phenomena. Their difference stems from the specific range of phoretic activity: lower activity prompts crystallization, while higher activity induces turbulence (SI Sec. III.F), corroborating our

predictions (Fig. 2). Ultimately, this discovery offers a paradigm to optimize the reconfigurability and functionality of active systems using minimal control.

The observed capability to discern both experimentally identified transitions is attributed to the simultaneous integration of long-range hydrodynamic and chemical interactions. Studies focusing solely on chemical interaction reproduced hexagonal crystallization but failed to capture active turbulence[44]. Conversely, simulations considering only hydrodynamic interaction illustrated turbulence but missed crystallization[80,81]. This comparative analysis highlights the unique predictive power unlocked by concurrently addressing both hydrodynamic and chemical interactions in modeling chemically active fluids—an aspect that merits broader recognition within the community.

## Methods

A more detailed description of the materials and methods is provided in SI Sec. II.

### Mathematical model and governing equations

We adopt the minimal physicochemical hydrodynamic model of IPA[55]. Without considering its internal flow, the IPA has been modeled as a

three-dimensional spherical particle or a 2D circular disk[32]. The Marangoni interfacial flow is represented by a solute-induced tangential velocity at the surface of particles or disks. The simple concept makes it a popular reference model for researching the behavior of a single IPA[32,34,35,57] or a pair[82,83]. Most importantly, it retains the critical signature of an IPA, viz., exploiting the convection of a chemical product to sustain a Marangoni propelling flow upon spontaneous symmetry-breaking. Concomitantly, the signature scenarios of IPAs observed in experiments[33,84] can be captured by the model[32,34,85]: by increasing Pe, it transitions from a stationary state to steady propulsion, and further to chaotic motion.

We describe the hydrochemical model by the dimensional Stokes equation for the velocity $\tilde{\mathbf{u}}$ and pressure $\tilde{p}$, along with the advection-diffusion counterpart for the solute concentration $\tilde{c}$ of a chemical species:

$$\tilde{\mathbf{\nabla}} \cdot \tilde{\mathbf{u}} = 0, \quad \tilde{\mathbf{\nabla}}\tilde{p} = \mu \tilde{\mathbf{\nabla}}^2 \tilde{\mathbf{u}}, \tag{3a}$$

$$\frac{\partial \tilde{c}}{\partial \tilde{t}} + \tilde{\mathbf{u}} \cdot \tilde{\mathbf{\nabla}}\tilde{c} = \mathcal{D}\tilde{\mathbf{\nabla}}^2 \tilde{c}, \tag{3b}$$

where $\mu$ is the dynamic viscosity of the solvent.

In addition to the fluid motion and solute transport, the model encodes the physicochemical activity at the disk surface by specifying its boundary conditions. This involves two processes: first, the uniform and constant solute emission from its surface

$$\mathcal{D}\mathbf{n} \cdot \tilde{\mathbf{\nabla}}\tilde{c} = -\mathcal{A}; \tag{4}$$

second, the generation of a slip velocity by a local solute gradient

$$\tilde{\mathbf{u}}_{\text{slip}} = \mathcal{M}(\mathbf{I} - \mathbf{nn}) \cdot \tilde{\mathbf{\nabla}}\tilde{c} \tag{5}$$

tangential to the surface.

The dimensionless equations are

$$\mathbf{\nabla} \cdot \mathbf{u} = 0, \quad \mathbf{\nabla}p = \mathbf{\nabla}^2\mathbf{u}, \tag{6a}$$

$$\frac{\partial c}{\partial t} + \mathbf{u} \cdot \mathbf{\nabla}c = \frac{1}{\text{Pe}}\mathbf{\nabla}^2 c. \tag{6b}$$

Here, we have chosen $a, \mathcal{AM}/\mathcal{D}, \mu\mathcal{AM}/(\mathcal{D}a)$, and $\mathcal{A}a/\mathcal{D}$ as the characteristic length, velocity, pressure, and concentration scales, respectively, e.g., $\mathbf{u} = \tilde{\mathbf{u}}/(\mathcal{AM}/\mathcal{D})$ and $c = \tilde{c}/(\mathcal{A}a/\mathcal{D})$. The dimensionless versions of (4) and (5) read

$$\mathbf{n} \cdot \mathbf{\nabla}c = -1, \tag{7a}$$

$$\mathbf{u}_{\text{slip}} = (\mathbf{I} - \mathbf{nn}) \cdot \mathbf{\nabla}c. \tag{7b}$$

The slip velocity Eq. (7b) allows the disk to freely move with a translational velocity $\mathbf{U}$ and a rotational velocity $\mathbf{\Omega} = \Omega\mathbf{e}_z$ with $\mathbf{e}_z = \mathbf{e}_x \times \mathbf{e}_y$. Note that (7b) is not the boundary condition for $\mathbf{u}$, which involves $\mathbf{U}$ and $\Omega$ as detailed in SI Sec. II.A.

It is worth noting that our numerical implementation does not exactly solve the Stokes equation (6a) but rather the Navier-Stokes equation featuring a finite yet small Reynolds number $\text{Re} = \mathcal{A}\mathcal{M}a/(\nu\mathcal{D})$, where $\nu$ is the kinematic viscosity of the solvent. We choose $\text{Re} = 0.5$ throughout this study. The associated inertia term is needed for time-marching the momentum equation. We find that its influence on the dynamics of a phoretic disk is reasonably weak, as discussed in SI Sec. II.D. In addition, we have also shown in SI Sec. III.F that variations in inertia within the range $\text{Re} \in [0.1, 2]$ do not

significantly alter the critical features of the collective phenomenon, including vortex size, velocity statistics, and the energy spectrum.

Unless otherwise specified, the size of the doubly-periodic domain is $L = 100$. In certain configurations, $L = 200$ and $L = 400$ are adopted.

## Numerical simulations

We numerically solve the dimensionless Eq. (6) in a periodic square domain of size $L$, subject to the boundary conditions (7) at the surfaces of freely moving disks. We have adapted a massively parallel flow solver[86,87] to cater for our physicochemical hydrodynamic configuration. The solver employs a lattice Boltzmann method to resolve fluid motion and solute transport while integrating an immersed boundary method to represent the surfaces of finite-sized disks. The implementation and validations are presented in SI Sec. II.

Using four computer nodes with four Intel Xeon Gold 6230R CPUs and 104 cores each, a typical simulation runs for several days to weeks, depending on Pe and $L$. For example, active turbulence tends to emerge at a high $\phi$ values, corresponding to a larger number of disks. Moreover, to broaden the energy spectrum of active turbulence, we use a domain of size $L = 200$ that doubles the default size, and even $L = 400$ for a more definitive insight (see Fig. 5j). These cases typically run for weeks using nine nodes to attain statistically invariant transitional or turbulent dynamics.

We have studied a few phoretic swimmers, as elaborated in SI Sec. III.D and illustrated through Supplementary Videos 9–11. Numerical simulations have successfully captured the key experimental observations on a pair of interacting active droplets[33].

## Data availability

The numerical source data[89] underlying Fig. 1d–f (top), 4a and 5a are provided with this paper. They are available at Zenodo (https://doi.org/10.5281/zenodo.7775033). The data shown in the figures and the corresponding MATLAB codes are shared in "Code Availability". Time-dependent field data, which are huge in size, are available upon request.

## Code availability

The MATLAB codes and related data[90] for reproducing the curve plots in the figures are available at https://github.com/qyang2025/phoretic_disks.git.

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

## Acknowledgements

The authors thank the helpful discussions with Ricard Alert, Cecile Cottin-Bizonne, Marjolein Dijkstra, Amin Doostmohammadi, Gerhard Gompper, Endao Han, Hisay Lama, Gaojin Li, Detlef Lohse, Sébastien Michelin, Alexander Morozov, Ran Ni, Ignacio Pagonabarraga, Fernando Peruani, Kai Qi, Francesc Sagués, Chunlei Song, Christophe Ybert, and Guangpu Zhu, as well as Wei-Fan Hu for sharing with us their data used in the validation. Q.Y. is supported by the research scholarship from the National University of Singapore and the China Scholarship Council. L.Z. acknowledges the Singapore Ministry of Education Academic Research Fund Tier 1 grant (A-8000197-01-00). Some computation of the work was performed on resources of the National Supercomputing Center, Singapore (https://www.nscc.sg), as well as the supercomputer Fugaku provided by RIKEN through the HPCI System Research Project (Project ID: hp230185).

## Author contributions

L.Z. designed and supervised the research. Q.Y., M.J. and L.Z. implemented the research. Q.Y., M.J., F.P. and L.Z. analyzed the data and wrote the manuscript.

## Competing interests

The authors declare no competing interests.
