## [Peer Review File · Nature Communications]

Reviewers' comments:

Reviewer #1 (Remarks to the Author):

In this manuscript the authors employ large-scale, agent-resolved simulations and demonstrate that modulating the activity of a wet phoretic medium yields a solid-liquid-gas phase transitions and a subsequent transition to a form of active turbulence in the fluid phase. It is claimed that several seemingly conflicting experimental observations on phoretic active systems are reproduced and reconciled by the simulations presented here providing eventually a unified landscape of their collective behaviors. The authors also argue that their findings enhance the understanding of long-range, many-body interactions among phoretic agents and offer new insights into non-equilibrium collective phenomena.

Before I go into the details, I like to offer a general assessment of the main claims and recommendation to the Editors of Nature Communications. Overall, I found the paper well written, interesting and presented in an accessible way. The paper offers indeed a nice survey of possible collective behaviors of self-phoretic active particles as a function of their density and activity. On the other hand, most of the findings have been reported already previously in other studies and it has not become sufficiently clear to me how the work here goes beyond the already established state of the art. In some place I also missed a more detailed comparison with previous results, for details see list given below. Regarding the comparison to experiments with camphor boats that the authors have singled out as the system closest to their model assumptions, the present work correctly reproduces the hexagonal lattice experimentally found in Ref. [38]. In contrast, the model here does not reproduce the "Kolmogorovian" active turbulence reported in Ref. [63]. The energy spectrum shown Fig. 5J does not display any notably power-law scaling (despite the indicated dashed lines) over at least one order of magnitude.

In summary, this paper is a solid and interesting piece of work reproducing many earlier results and connecting them together nicely in a phase diagram. There are some novel aspects such as the elegant approximate analytical argument for the solid-liquid transition in Eq. (2) and the coexistence between high-density regions of vortices and clusters with dilute gas-like phases at high activity. It does, however, not really live up to the claims of the abstract regarding reconciling conflicting experiments, enhancing the understanding of long-range interactions and giving new insights into non-equilibrium phenomena. Altogether, I therefore do not recommend publication in Nature Communications, where articles should provide novel insights for a wider general audience. The paper is, however, suitable for publication in a good, more specialized journal.

List of detailed points:

1. Solid-liquid transitions: The hexagonal arrangement of active particles was first reported in [1] and analyzed in detail in Ref. 61 of the current manuscript. I didn't see much difference in the findings reported here beyond the different mechanism leading to active particle motion (self-propulsion vs. autophoresis).

2. Active turbulence: the comparison to the well-studied cases of active turbulence in active nematics and polar active fluids such as bacterial suspensions, see e. g. Ref. 13 in the manuscript or [2] for recent reviews, are not very convincing. I was puzzled by the comparison of the spectrum in Fig. 5 with the well-known spectrum of active nematics, because the present work deals with isotropic, circular active particles, whereas active nematics are usually elongated particles (biopolymers activated by molecular motors or rod-shaped swimming bacteria). As already mentioned above, the comparison with the power-law portions indicated by dashed lines was not very convincing anyway (the author may want to remove the dashed lines). The spectrum here rather does not display any notable power-law dependence.

3. Cluster analysis: The exponents of the power-law fit in Fig. 4G are not given explicitly. How does the clustering reported here compare to the one found earlier in simulations (see [3]) and experiments of interacting autophoretic particles ?

4. In the region termed as "active turbulence" here, the authors report formation of irregular chain of particles. How do these chains compare to the ones found earlier in colloidal particles with dipolar interactions, see [4-6].

5. Comparison to experiments: The authors singled out Refs. [38] and [63] for the comparison of their simulation results with the vast experimental literature on collective behavior of active colloids. Are there other systems in soft matter or in living active systems than camphor boats that are potentially described by the presented model ?

Additional references:

[1] Crystallization in a dense suspension of self-propelled particles

J Bialké, T Speck, H Löwen - Physical review letters, 2012

[2] Emergence of active turbulence in microswimmer suspensions due to active hydrodynamic stress and volume exclusion

K Qi, E Westphal, G Gompper, RG Winkler - Communications Physics, 2022 - nature.com

[3] Dynamic clustering and chemotactic collapse of self-phoretic active particles

O Pohl, H Stark - Physical review letters, 2014

[4] Phase diagram of two-dimensional systems of dipole-like colloids

H Schmidle, CK Hall, OD Velev, SHL Klapp - Soft Matter, 2012 - pubs.rsc.org

[5] Collective dynamics of dipolar and multipolar colloids: From passive to active systems

SHL Klapp - Current opinion in colloid & interface science, 2016 – Elsevier

[6] Emergent behavior in active colloids

A Zöttl, H Stark - Journal of Physics: Condensed Matter, 2016 - iopscience.iop.org

Reviewer #2 (Remarks to the Author):

Qianhong Yang et. al. studied the collective behavior of isotropic phoretic agents. Using large-scale computer simulations, the authors demonstrated solid-liquid-gas phase transitions and also laminar-turbulent transitions in the active system's fluid phase. This study, I believe, is of significance to the scientific community by providing a paradigmatic framework to advance our understanding of the phase behavior of active suspension, tunable control of active matter and non-equilibrium phenomena in general.

The authors presentation is succinct and clear. The manuscript is suitable for publication in Nature communications. However, while the premise of this study is of significant importance, I have some comments about the technical details of the manuscript which I would like to have the authors' response:

Major

1.) The individual active particles' linear translation velocity (U) is central to the analyses of this manuscript. However, the authors do not clearly explained how they calculated the translation (U) and rotational (Ω) velocities neither in the manuscripts' main text nor in the supporting documentation. The only reference to the translation and rotational velocities relationship with the total force (and torque) acting on the particles is in equations (19a,b) of the Supplemental Information (SI).^{[1][2][3][4][5][6][7][8][9][10][11][12][13][14][15][16][17][18][19][20][21][22][23][24][25][26][27][28][29][30][31][32][33][34][35][36][37][38][39][40][41][42][43][44][45][46][47][48][49][50][51][52][53][54][55][56][57][58][59][60][61][62][63][64][65][66][67][68][69][70][71][72][73][74][75][76][77][78][79][80][81][82][83][84][85][86][87][88][89][90][91][92][93][94][95][96][97][98][99][100]}

This is perhaps inadequate. U and Ω are unknowns that would only be fixed by imposing the net zero force ($F=0$) and torque ($L=0$) constraints. The distinguishing characteristic of active phoretic particles is that the total force (and torque) acting on the phoretic particle (particle + thin/interfacial layer) is zero. That is, $F=0$ and $L=0$ irrespective of whether Reynolds number (Re) vanishes or not. $F=0$ and $L=0$ are the necessary conditions that would allow us to uniquely determine the particles' translation and rotational velocities (U and Ω). [see eqns. 3 and 4 in review article <https://doi.org/10.1146/annurev-fluid-120720-012204>].^{[1][2][3][4][5][6][7][8][9][10][11][12][13][14][15][16][17][18][19][20][21][22][23][24][25][26][27][28][29][30][31][32][33][34][35][36][37][38][39][40][41][42][43][44][45][46][47][48][49][50][51][52][53][54][55][56][57][58][59][60][61][62][63][64][65][66][67][68][69][70][71][72][73][74][75][76][77][78][79][80][81][82][83][84][85][86][87][88][89][90][91][92][93][94][95][96][97][98][99][100]}

2.) Following #1, we also expect full Eulerian description of the dynamics with the velocity U and Ω included in the flow boundary conditions (see eqns. 7a,b of the main text and 18a,b of the SI). Why are U and Ω missing in the boundary conditions?^{[1][2][3][4][5][6][7][8][9][10][11][12][13][14][15][16][17][18][19][20][21][22][23][24][25][26][27][28][29][30][31][32][33][34][35][36][37][38][39][40][41][42][43][44][45][46][47][48][49][50][51][52][53][54][55][56][57][58][59][60][61][62][63][64][65][66][67][68][69][70][71][72][73][74][75][76][77][78][79][80][81][82][83][84][85][86][87][88][89][90][91][92][93][94][95][96][97][98][99][100]}

3.) I also find it strange the assertion that the gradient of the solute concentration vanish ($\text{grad } c = 0$) mid-way between two particles due to hexagonal symmetry (see the text below eqn. 1 of the main text). The solute concentration contribution from each individual particle is a long-range $1/r$ term similar to electrostatic potential with a gradient decay ($\sim 1/r^2$). Can the authors justify this assertion?

4.) Distinct from the instability induced propulsion central to the manuscript discussion, two or more chemically active particles with non-zero mobility (M) would always move with finite translation velocity (U) and zero rotational velocity ($\Omega=0$) due to the spherical symmetry of the particles. This effect is always present irrespective of the value of the Peclet number (Pe). This seems to be completely missing in the manuscripts' discussion.

Minor

5.) I find the use of the word “droplets”, while ignoring the particles' internal fluid flow, to describe these active particles throughout this manuscript unjustified. Yes, if the capillary number is small and the viscosity of the inner fluid of the “droplet” is high, the authors can justify ignoring the inner fluid flow. However, in that limit (which I presume the authors imply), calling the particles “droplets” could be misleading. The authors could simply maintain the name “isotropic phoretic agents (IPA)” for the chemically active phoretic particles initially introduced in the main text.

Response to Reviewer I regarding manuscript titled “Shaping active matter: from crystalline solids to active turbulence” authored by Qianhong Yang, Maoqiang Jiang, Francesco Picano, and Lailai Zhu, submitted to Nature Communications

Referee 1: *In this manuscript the authors employ large-scale, agent-resolved simulations and demonstrate that modulating the activity of a wet phoretic medium yields a solid-liquid-gas phase transitions and a subsequent transition to a form of active turbulence in the fluid phase. It is claimed that several seemingly conflicting experimental observations on phoretic active systems are reproduced and reconciled by the simulations presented here providing eventually a unified landscape of their collective behaviors. The authors also argue that their findings enhance the understanding of long-range, many-body interactions among phoretic agents and offer new insights into non-equilibrium collective phenomena.*

Our response 1: We sincerely thank the reviewer for their meticulous evaluation of our manuscript and their insightful feedback. In the subsequent sections, we make a concerted effort to address the concerns raised, and implement revisions (marked in magenta) in the manuscript where necessary. We are confident that these adjustments will significantly enhance the clarity and overall quality of our work. Moreover, we have underscored certain parts of the response to facilitate a more streamlined reading experience.

Referee 2: *Before I go into the details, I like to offer a general assessment of the main claims and recommendation to the Editors of Nature Communications. Overall, I found the paper well written, interesting and presented in an accessible way. The paper offers indeed a nice survey of possible collective behaviors of self-phoretic active particles as a function of their density and activity. On the other hand, most of the findings have been reported already previously in other studies and it has not become sufficiently clear to me how the work here goes beyond the already established start of the art.*

Our response 2: We acknowledge the reviewer’s positive appraisal of our work as ‘well written, interesting and presented in an accessible way’. We agree that at first glance, some of our findings appear to align with those previously reported, as we have cited if not unintentionally missed. On the other hand, we think there may have been a misunderstanding regarding the novelty of our work. We will take this opportunity to spotlight our distinctive contributions that extend beyond the current state of the art.

Contribution I: We have, for the first time, demonstrated the coexistence of phase transition and laminar-turbulent transition, which are induced by progressively increasing the activity—Péclet (Pe) number of our active matter. No similar coexistence of both transitions has been reported in previous works to the best of our knowledge. This finding has been highlighted in both the abstract and conclusion of our original manuscript.

The observed progressive development of phase transition and laminar-turbulent transition carries significant implications, suggesting a closer similarity between active matter and classical matter than previously recognized. Namely, we can control this progression through all traditional thermodynamic phases by manipulating the activity level, which functions as an effective temperature. Once in the fluid phase, we can further steer this active fluid’s lam-

inar, transitional, and turbulent states by tuning the same activity level, which now behaves like an effective Reynolds number.

Notably, this remarkable observation reconciles two apparently contradictory experiments (Soh et al., 2008; Bourgoïn et al., 2020) involving camphor surfers: the first (Soh et al., 2008) presents the formation of a hexagonal crystal (indicating phase transition), and the second (Bourgoïn et al., 2020) reveals the emergence of active turbulence (via laminar-turbulent transition). However, neither of these studies, nor any other, have reproduced both transitions simultaneously, leaving the apparent contradiction unresolved. Our research uniquely reproduces the Pe-dependent transitions that occur progressively, attributing their divergence to the distinct parameter (Pe) spaces they inhabit—lower Pe for crystallization and higher Pe for turbulent dynamics. More details regarding the reconciliation will be provided in Our response 5.

Contribution II: The success of simultaneously capturing both transitions stems from the significant technical advancements made in our work. These advancements pertain specifically to addressing both chemical and hydrodynamic interactions between phoretic agents. When considering the chemical interaction alone, hexagonal crystallization can be reproduced, but not active turbulence (Gouiller et al., 2021). The authors attribute their inability to capture turbulence to the omission of hydrodynamic interactions between swimmers. Conversely, focusing solely on hydrodynamic interaction may lead to turbulence but no crystallization (Zantop and Stark, 2022; Qi et al., 2022). As one of the rare attempts that address both interactions, we demonstrate the criticality of considering such long-range interactions. This consideration has indeed presented a persistent challenge and area of concern for modeling active fluids (Cates and Tailleur, 2015; Liebchen and Mukhopadhyay, 2021; Zöttl and Stark, 2023), quoted below:

“... , hydrodynamic interactions could be responsible for the arrest of coarsening or, if we are unlucky, they could destroy MIPS entirely” (Cates and Tailleur, 2015)

... , systems of $N \gg 10^3$ active colloids with phoretic, osmotic and hydrodynamic near-field interactions are still hard to simulate with state-of-the art simulation techniques. However, simulations of such system sizes would be highly desirable e.g. to understand the large-scale collective behaviour of active colloids in three dimensions which has been hardly explored so far. (Liebchen and Mukhopadhyay, 2021)

Explicit modeling of hydrodynamic and phoretic fields is necessary for dense systems and to access the full hydrodynamic-chemical coupling. (Zöttl and Stark, 2023)

Despite these hypothetical or suggestive arguments, we assert that our study might be the first to provide a concrete illustration demonstrating the distinctive predictive capabilities unlocked by factoring in both long-range hydrodynamic and chemical interactions.

Contribution III: Our study uncovers another crucial finding—the active fluid follows a unique path toward a turbulent-like state. The detected pathway shares similarities with the development of inertial turbulence from a laminar flow as the Reynolds number increases,

encompassing instability, secondary instability, and a subsequent transition to turbulence. To the best of our knowledge, this is the inaugural report that documents such a hydrodynamic route to turbulence for agent-resolved active fluids, unveiling a closer phenomenological resemblance between active and classical fluids in their laminar-turbulent transition than previously demonstrated.

Considering the above-highlighted novelties beyond the current state of the art, we kindly request the reviewer to reconsider their assessment of our work.

Referee 3: *In some place I also missed a more detailed comparison with previous results, for details see list given below.*

Our response 3: We acknowledge that there were a few we inadvertently missed, as the reviewer rightly pointed out. Please rest assured that this was not intentional. It can be challenging to capture every pertinent reference given the expansive and rapidly evolving literature in the field of active matter. We sincerely thank the reviewer for bringing these to our attention, which have been aptly discussed in the revised manuscript and/or SI when necessary. In the following responses, we will compare our findings with the previous results in detail.

Referee 4: *Regarding the comparison to experiments with camphor boats that the authors have singled out as the system closed to their model assumptions, the present work correctly reproduces the hexagonal lattice experimentally found in Ref. [38]. In contrast, the model here does in my reproduce not reproduce the “Kolmogorovia” active turbulence reported in Ref. [63]. The energy spectrum shown Fig. 5J does not display any notably power-law scaling (despite the indicated dashed lines) over at least on order of magnitude.*

Our response 4: We would like to argue that we did not ‘single out camphor boats as the system close to our model assumption’. We provide a detailed clarification in Our Response 10, in response to a similar comment from the reviewer.

We appreciate the reviewer’s comments and would like to clarify certain points. We respectfully disagree with the assertion that we claimed to reproduce the “Kolmogorovia” active turbulence reported in Ref. [63] (Bourgoin et al., 2020). A thorough review of our original manuscript and SI shows no reference to “Kolmogoro”, “Kolmogrovi”, or “Kolmogrovia”-related terminology. These terms only appear in the titles of the references cited, leading to some confusion on our part.

In the original submission, the only statement relevant to the reviewer’s comment is, “Our work reproduces and reconciles several seemingly conflicting experimental observations on phoretic active systems”. We did not mean to imply the reproduction of Kolmogorovian turbulence characterized by an energy spectrum $E(k) \sim k^{-5/3}$ in the high wavenumber k regime, because our energy spectrum does not reflect such a scaling.

We further acknowledge the constructive critique concerning power-law scaling. Upon re-visiting our results, we concede that the initially identified scaling $E(q) \sim q^{-4}$ may have been

Figure 1: Power-law scaling range of energy spectra as presented in well-recognized experimental (A-F) and numerical (G-J) studies on active turbulence. A, suspension of *B. subtilis*, adapted from Wensink et al. (2012). B, suspension of ram spermatozoa, adapted from Creppy et al. (2015). C, electrically driven Janus colloids in a monolayer, adapted from Nishiguchi and Sano (2015). D, three-dimensional suspension of *E. coli*, adapted from Liu et al. (2021). E, monolayers of different tissue cells, adapted from Lin et al. (2021). F, a thin film of microtubules, adapted from Martínez-Prat et al. (2021). G, two-dimensional active nematodynamics, adapted from Giomi (2015). H, three-dimensional active nematodynamics, adapted from Urzay et al. (2017). I, a monolayer suspension of prolate squirmers confined between two plates, adapted from Qi et al. (2022). J, a monolayer of squirming rods within two plates, adapted from Zantop and Stark (2022).

ambiguous, and thus, it has been removed from our revised manuscript. More importantly, we have conducted a simulation with a doubled domain size $L = 400$ (compared to the original $L = 200$) for the data point $(\phi, \text{Pe}) = (0.5, 20)$ shown in Fig. 5J of the initial manuscript. Using the new data, we demonstrate a more pronounced power-scaling $E(q) \sim q^1$ in the small q (large length scale) regime, as shown in Fig. 5J of the revised manuscript.

On the other hand, we respectfully disagree with the notion that active turbulence must demonstrate a notable power-law scaling in the energy spectrum over at least one order of magnitude, as appears to be implied by the reviewer's comments (assuming our understanding is correct). In support of our stance, we have meticulously reviewed and summarized

energy spectra from well-recognized experimental (Wensink et al., 2012; Creppy et al., 2015; Nishiguchi and Sano, 2015; Liu et al., 2021; Lin et al., 2021; Martínez-Prat et al., 2021) and numerical (Giomi, 2015; Urzay et al., 2017; Qi et al., 2022; Zantop and Stark, 2022) studies on active turbulence. This is illustrated in Figure. 1.

We draw attention to the above-mentioned six experimental datasets, selected as representative examples by the recent review titled “Active Turbulence”. This review, by Alert et al. (2022), was published in the prestigious journal “Annual Review of Condensed Matter Physic”. Among the four numerical studies, works by Qi et al. (2022) and Zantop and Stark (2022), from the research groups of Prof. Gerhard Gompper and Prof. Holger Stark, respectively, are particularly notable. The reviewer has referred to their works (Refs. [2], [3], and [6] within the reviewer’s report) to support his/her viewpoints in other contexts, indicating a certain regard for their researches.

Upon careful scrutiny of the power-law scaling in these ten collected studies, we found that none of the scalings spans over an order of magnitude. The widest span identified here possibly corresponds to the scaling $E(k) \sim k^{5/3}$, apparent in the orange and red lines in Figure. 1H around $k/k_c = 10^{-1}$ (Qi et al., 2022), and the scaling $E(k) \sim k^{1.4}$, represented by the blue line in Figure. 1I around $k/k_c = 10^{-1}$ (Zantop and Stark, 2022). Even though, both scalings cover approximately 0.6 – 0.7 order of magnitude, while the other presented scalings typically cross a range of 0.2 – 0.3 order of magnitude.

In light of these observations about the power-law scaling of energy spectra in previous well-regarded works, we kindly ask the reviewer to reassess the new scaling presented in Fig. 5J of our revised manuscript.

Referee 5: *In summary, this paper is a solid and interesting piece of work reproducing many earlier results and connecting them together nicely in a phase diagram. There are some novel aspects such as the elegant approximate analytical argument for the solid-liquid transition in Eq. (2) and the coexistence between high-density regions of vortices and clusters with dilute gas-like phases at high activity. It does, however, not really live up to the claims of the abstract regarding reconciling conflicting experiments, enhancing the understanding of long-range interactions and giving new insights into non-equilibrium phenomena. Altogether, I therefore do not recommend publication in Nature Communications, where articles should provide novel insights for a wider general audience. The paper is, however, suitable for publication a good, more specialized journal.*

Our response 5: We appreciate the reviewer for acknowledging our work as solid an interesting. We also thank the reviewer’s recognition of certain novel aspects of our work—the approximate analytical argument and the coexistence of vortices and clustering. However, we believe that these aspects, while important, may not encapsulate the most significant contributions of our work. These have been emphasized in Our response 2.

As the reviewer has expressed reservations regarding our assertion of ‘reconciling conflicting experiments, enhancing the understanding of long-range interactions, and giving new insights into non-equilibrium phenomena’, we see this as an opportunity to clarify and provide

further evidence in support of our claims. We delve into this substantiation below.

- **Reconciling conflicting experiments:** As described in Our response 2, this argument refers to two apparently contradictory experiments (Soh et al., 2008; Bourgoïn et al., 2020) on camphor surfers: the first (Soh et al., 2008) exhibits the formation of a hexagonal crystal, and the second (Bourgoïn et al., 2020) reveals the emergence of active turbulence. However, neither of these studies, nor any other, have reproduced both phenomena simultaneously, leaving the apparent contradiction unresolved.

We reproduce both concurrently, attributing their divergence to the distinct parameter (Pe) spaces they inhabit—lower Pe for crystallization and higher Pe for turbulence. This difference in Pe is largely due to the variation in the size of the camphor disks used: the disk radius is 0.5 mm in Soh et al. (2008), whereas it is 2.5 mm in Bourgoïn et al. (2020). Moreover, the disk height is quite similar in both studies, specifically 500 μm and 600 μm , respectively. Furthermore, both experiments followed the same protocol (Campbell et al., 2004; Smoukov et al., 2005) to fabricate the disks, which should ensure their comparable levels of chemical activity. Besides the disk itself, another factor that can influence its behavior is the depth of the subsurface fluid, which was set at 5 mm in Soh et al. (2008) and at 10 mm in Bourgoïn et al. (2020). However, in both scenarios, these depths are considerably greater than three times the radius of the disk. As per the study by Boniface et al. (2019), when the depth of the fluid exceeds this threshold, its influence on the disk’s behavior becomes negligible. Collectively recalling the linear dependence of Pe on the disk radius, we deduce that the Pe of the latter experiment is approximately five times that of the former one, a trend that indeed qualitatively aligns with our predictions. The discussion on the two experiments has been added in the updated SI.

- **Enhancing the understanding of long-range interactions:** We would like to emphasize that our original sentence was ‘enhance the understanding of long-range, many-body interactions among phoretic agents’, where the term ‘among phoretic agents’ precisely narrows down the context of the statement.

Besides, we would like to reiterate, as mentioned in Our response 2, that fully considering both long-range hydrodynamic and chemotactic interactions between phoretic microswimmers represents a substantial challenge in the modeling of active fluids (Cates and Tailleur, 2015; Liebchen and Mukhopadhyay, 2021; Zöttl and Stark, 2023). Prevailing studies often limit their focus to dry active matter, namely, neglecting hydrodynamic effects and consequently failing to conserve the momentum of the active system. A minority of research incorporates either hydrodynamic or chemical interactions, but seldom both.

Our integration of both interactions enables us to numerically capture both experimental phenomena—crystal formation (Soh et al., 2008) and active turbulence (Bourgoïn et al., 2020)—accomplishments that, to the best of our knowledge, are unprecedented. More specifically, by progressively adjusting the activity Pe, we successfully observe a phase transition at lower Pe, followed by a laminar-turbulent transition at higher Pe.

We credit this successful capture of both transitions in our simulation to the distinctive integration of long-range hydrodynamic and chemical interactions. This assertion

is reinforced by the comparative analysis of our simulation with those considering either hydrodynamic or chemical interaction. Studies focusing solely on chemical interaction can reproduce hexagonal crystallization but fail to capture active turbulence (Gouiller et al., 2021). Conversely, simulations considering only hydrodynamic interaction may lead to turbulence, but not crystallization (Zantop and Stark, 2022; Qi et al., 2022). This stark contrast further emphasizes the indispensability of accounting for both hydrodynamic and chemical interactions in modelling active fluids, a fact that may not be fully recognized by the community.

Furthermore, our study unveils how the interplay between chemical repulsion and hydrodynamic attraction manifests different collective behaviors. The chemical repulsion causes the formation of Wigner crystal, a structure significantly distinct from the crystalline formations observed in prior microswimmer studies (Bialké et al., 2012; Klamser et al., 2018), as the reviewer also noted. For a detailed discussion on this difference, we refer to the section Our response 6 regarding the reviewer’s pertinent comment. We have developed an ‘elegant approximate analytical argument’ (as commended by the reviewer) to predict the threshold for the formation of these chemical-repulsion-induced Wigner crystals. We believe this prediction betters the understanding of chemical repulsion between phoretic agents. We also carefully characterize, in Section IV and V of the original SI, the hydrodynamic attraction among the phoretic agents, resultant of their dipolar flow pattern (insightfully recognized by the reviewer) characteristic of pusher swimmers. We have shown that the hydrodynamic attraction intensifies with the phoretic activity (Pe), elucidating the observed dynamic chaining of phoretic swimmers at higher Pe values, and the qualitatively similar chain formation in active droplet experiments (Thutupalli et al., 2011, 2018).

Taken together, we argue that our work meaningfully advances our understanding of the long-range interactions among phoretic active colloids.

- **Giving new insights into non-equilibrium phenomena:** Contrary to the reviewer’s interpretation, our initial phrase was ‘offer new insights into non-equilibrium collective phenomena’, with the term ‘collective’ specifically refining the context. To further improve precision and narrow down the scope, we have revised ‘collective phenomena’ to ‘their collective behaviors’, where ‘their’ refers to the phoretic agents mentioned in the preceding sentence. Thus, the new statement now reads ‘These findings enhance the understanding of long-range, many-body interactions among phoretic agents, offer new insights into their non-equilibrium collective behaviors’.

We subsequently evaluate the suitability of this updated statement, namely, ‘offer new insights into their non-equilibrium collective behaviors’. Firstly, we study collective behavior of phoretic active agents that swim by continuously consuming chemical energy, thereby indeed displaying non-equilibrium features. Secondly, the new insights pertain to the unique contributions our research offers, which may have been underappreciated by the reviewer, and are emphasized in Our response 2. Briefly, they include: 1) the unprecedented reproduction of progressively coexisting phase transition and laminar-turbulent transition; 2) the demonstration of the indispensable role both hydrodynamic and chemical interactions play in the non-equilibrium collective behaviors; 3) the identification of a unique hydrodynamic path that active fluids follow en route to turbulent-like collective states.

Moreover, we acknowledge the reviewer’s concern about the suitability of our work ‘for a wider general audience’. However, we contend that our research, being highly interdisciplinary, indeed has the potential to appeal to a diverse general audience. The relevance of our work across multiple fields is elucidated as follows:

- **Physical chemistry:** Our work draws inspiration from chemically active droplets and camphor surfers, concepts well-acknowledged within the physical chemistry community (Nakata et al., 1997; Soh et al., 2008; Banno et al., 2012; Meredith et al., 2020; Kichatov et al., 2021; Meredith et al., 2022; Matsuo et al., 2023).
- **Fluid mechanics and transport phenomena:** The relevance is two-fold. Firstly, we start from the collective motion of interacting phoretic disks through the lens of low-Reynolds-number hydrodynamics (Happel and Brenner, 1983) with physicochemical effects (Lohse and Zhang, 2020). Secondly, the delineated hydrodynamic path to active turbulence is likely to captivate the research community engaged in the study of hydrodynamic stability, transition, and turbulence of classical inertial flows (Schmid and Henningson, 2002).
- **Thermodynamics and statistical mechanics:** Our studied active matter undergoes a phase transition, revealing all thermodynamic phases dependent on the activity that is analogous to temperature. Specifically, we identify defect-mediated melting in a two-dimensional active material through a hexatic phase, a phenomenon aptly described by the renowned KTHNY theory (Kosterlitz and Thouless, 1973; Halperin and Nelson, 1978; Young, 1979).
- **Biophysics:** The plethora of emergent collective patterns exhibited by our studied phoretic agents mirror those observed in biological systems, e.g., cellular crystallization (Petroff et al., 2015; Tan et al., 2022), epithelial-mesenchymal melting transition (Jordan et al., 2011; Pasupalak et al., 2020), and bacterial turbulence (Wensink et al., 2012; Peng et al., 2021). This correlation could provide not only novel insights into the mechanisms underlying active biological collectives but could also inspire the design of bio-mimetic synthetic active materials (Palacci et al., 2013; Needleman and Dogic, 2017).
- **Scientific computing:** We adapt an in-house, massively parallel, interface-resolved flow solver integrating Lattice Boltzmann method and immersed boundary method (Jiang and Liu, 2019; Jiang et al., 2022) to address a physicochemical hydrodynamic problem, successfully demonstrating solid validations against published theoretical and numerical data (Hu et al., 2019). These instances could potentially serve as benchmark computational studies in this emerging, highly interdisciplinary field of physicochemical hydrodynamics (Lohse and Zhang, 2020).

In interim summary, we have tried our best to address the reviewer’s concerns related to the ‘the claims of the abstract’ and the provision of ‘novel insights for a wider general audience’. We respectfully request the reviewer to reconsider their initial recommendation in light of this response and the corresponding amendments.

Referee 6: *Solid-liquid transitions: The hexagonal arrangement of active particles was first reported in [1] and analyzed in detail in Ref. 61 of the current manuscript. I didn’t see*

much difference in the findings reported here beyond the different mechanism leading to active particle motion (self-propulsion vs. autophoresis).

[1] Crystallization in a dense suspension of self-propelled particles
 J Bialké, T Speck, H Löwen - Physical Review Letters, 2012

Our response 6: We appreciate the reviewer for referencing [1] (Bialké et al., 2012). This seminal work, which numerically identifies the hexagonal state of dry active particles, is indeed an important contribution in the field of active matter. We regret that it was initially overlooked in our study. To improve the completeness of our work, we have included [1] in our updated manuscript. We would also like to underscore that, our initial manuscript (left bottom paragraph on page two) does highlight the difference between the crystalline structure we observe and the other structures reported in [1] and Ref. 61 (Klamser et al., 2018) suggested by the reviewer.

Beyond the mechanism underlying the crystal formation mentioned by the reviewer, the most significant distinction lies in the packing fraction (or packing efficiency) of the crystal, as elaborated below.

Figure 2: Hexagonal closed-packed crystal (upper row) and hexagonal Wigner crystal (lower row) formed in active matter systems. A, Brownian dynamics computer simulations of a Yukawa model of self-propelled particles, adapted from Bialké et al. (2012). B, a monolayer of photoactivated Janus colloids, adapted from Palacci et al. (2013). C, bacterial crystal formed by *Thiovulum majus*, adapted from Petroff et al. (2015). D, Wigner crystal-like lattice formed by camphor surfers (Soh et al., 2008). E, Wigner crystal of isotropic phoretic agents we observe (taken from Fig. 1D of the original manuscript), with an area fraction of $\phi = 0.12$.

In the two referred works and many others (Palacci et al., 2013; Petroff et al., 2015; Briand and Dauchot, 2016; Singh and Adhikari, 2016; Kichatov et al., 2021), the crystal presents a monolayer hexagonal close-packed (HCP) arrangement (see the upper row of Figure 2); in

this setting, the swimmers representing atoms pack as densely as possible, hence approaching the maximum possible packing fraction (PF) of $\pi / (2\sqrt{3}) \approx 0.907$ (Kittel and McEuen, 2018; West, 2022)—not to be confused with the well-known PF of $\pi / (3\sqrt{2}) \approx 0.74$ for the more general three-dimensional HCP arrangement: A graphene layer typically features $\text{PF} \approx 0.907$, whereas metals like magnesium and titanium crystallize in the HCP structure with $\text{PF} \approx 0.74$.

In contrast, the packing fraction—mirroring the area fraction ϕ we have defined—of our examined crystal structure can plummet to as low as 0.005, which corresponds to the data point at the lower left corner in Fig. 2 of our manuscript. Such a crystalline configuration embodies what is known as the Wigner crystal, exemplified by the lower row of Figure. 2. Theoretically predicted by Eugene Wigner in 1934 (Wigner, 1934, 1938), this solid phase of electrons was only recently visualized in experiments (Li et al., 2021). A original Wigner crystal forms at low electron densities, where the distance between electrons significantly exceeds the effective “electron size”, denoted by the wave function’s extent. The separation of these two length scales enables long-range Coulomb repulsion between electrons to overpower their kinetic energy, thus giving rise to an ordered lattice structure.

The formation of a Wigner crystal from our phoretic disks can be traced back to the established equivalence between phoretic and Coulomb interactions (Illien et al., 2017; Golestanian, 2019; Liebchen and Mukhopadhyay, 2021): both exhibit repulsive characteristics and their potential decays slowly as $1/r$ with r being the distance. Drawing parallels with the conditions for the formation of the original Wigner crystal—where the electrons’ kinetic energy is significantly weaker than their repulsion—in our case of phoretic agents, they crystallize when their activity, indicated by the Péclet (Pe) number, falls below a certain threshold (refer to Fig. 2 in our manuscript).

We wish to emphasize that our initial manuscript succinctly highlighted the aforementioned distinction, as quoted verbatim below:

“... implies that this solid shares similarity with the Wigner crystal constituting electrons, as theoretically predicted [35] and directly visualized in experiments very recently [36]. Unlike the long-range Coulomb force causing the electronic Wigner crystallization, the chemorepulsion among phoretic disks [17, 23, 37] creates the active Wigner crystal. This phenomenon agrees with the experimental observations on camphor sulfurs [38] and active droplets [18], as well as numerical prediction for the former [39]. It differs from the closely-packed crystallization in other active suspensions [40–43] that mimic canonical crystalline solids.”

To further accentuate this contrast, we have modestly expanded the discussion in the updated SI.

Referee 7: *Active turbulence: the comparison to the well-studied cases of active turbulence in active nematics and polar active fluids such as bacterial suspensions, see e. g. Ref. 13 in the manuscript or [2] for recent reviews, are not very convincing. I was puzzled by the comparison of the spectrum in Fig. 5 with the well-known spectrum of active nematics, because the present work deals with isotropic, circular active particles, whereas active nematics are usually elongated particles (biopolymers activated by molecular motors or rod-shaped*

swimming bacteria). As already mentioned above, the comparison with the power-law portions indicated by dashed lines was not very convincing anyway (the author may want to remove the dashed lines). The spectrum here rather does not display any notable power-law dependence.

[2] Emergence of active turbulence in microswimmer suspensions due to active hydrodynamic stress and volume exclusion

K Qi, E Westphal, G Gompper, RG Winkler - Communications Physics, 2022 - nature.com

Our response 7: We would argue that the new energy spectrum based on the simulation with a larger domain has shown an evident power-law scaling $E(q) \sim q$ in the low q regime, see Fig. 5J of the revised manuscript.

We admit that we compared our data with the previous studies on active nematic turbulence. However, we would like to underscore that the comparison with the study on polar active fluids, Ref. [13] (Giomi and Marchetti, 2012), was not framed within the context of active turbulence as the reviewer suggested. A thorough examination of Ref. [13] reveals that the term ‘turbulence’ does not appear. That study focused on the pattern formation of two-dimensional active nematics undergoing instability and transition. Indeed, we referred to Ref. [13] as it illustrates the oscillatory instability of active fluids that mirrors the classical hydrodynamic instability scenario, which is consistent with our observations on oscillatory instability (see Fig. 4 of our manuscript). We would like to maintain that there is a significant difference between instability, transition, and turbulence.

We concur with the reviewer that drawing comparisons between our energy spectrum and that of active nematic turbulence (Alert et al., 2020; Martínez-Prat et al., 2021) may not be entirely justified. To avoid misleading readers, we indeed originally highlighted the differences between the two systems in the previous manuscript, as quoted below:

“This trend imitates the universal scaling of 2D active nematic turbulence obtained theoretically and experimentally [58–60], despite the disparity between the two systems. ”

As discussed in Our response 4, we have eliminated the power-law scaling $E(q) \sim q^{-4}$ at the large q regime in the revised manuscript. Consequently, the resemblance between our energy spectrum and that of active nematic turbulence is further lessened.

In response to the reviewer’s comment, we wish to discuss the potential reasons behind the observed similarities between the suspensions of isotropic phoretic agents (IPAs) and active nematics or polar active fluids in the context of active turbulence. The recent review by Alert et al. (2022) categorizes active turbulence into two types, based on whether the contributing entities exhibit polar or nematic behaviors, indeed as the reviewer wisely pointed out. Interestingly, we find that the active turbulence of IPAs, which we study, might represent a third category. This new category demonstrates distinct characteristics from both categories—it shares the unique oscillatory transitional scenario with polar active fluids (Giomi and Marchetti, 2012; Alert et al., 2022) and exhibits the scaling of the energy spectrum typical of active nematics (Alert et al., 2020; Martínez-Prat et al., 2021). A simple, though perhaps superficial, explanation for these observations could be attributed to the

dual-state nature of an IPA. In its stable state, an IPA is stationary and thus apolar, similar to a nematic entity. However, when unstable, it behaves like a polar swimmer, akin to a Janus colloid or bacterium.

We trust that the aforementioned explanation may help to alleviate some of the reviewer’s confusion. We have also included it in the revised SI.

Referee 8: *Cluster analysis: The exponents of the power-law fit in Fig. 4G are not given explicitly. How does the clustering reported here compare to the one found earlier in simulations (see [3]) and experiments of interacting autophoretic particles ?*

[3] Dynamic clustering and chemotactic collapse of self-phoretic active particles
O Pohl, H Stark - Physical Review Letters, 2014

Our response 8: We thank the reviewer for directing us to [3] (Pohl and Stark, 2014). Using Brownian dynamics simulations, the authors studied dilute suspensions of Janus-type phoretic colloids driven by their self-generated chemical fields, undergoing translational and rotational motion. Importantly, the study reproduces the experimentally observed dynamic clustering of Janus colloids, revealing significant clustering when the two types of motion lead to competing attractive and repulsive interactions, respectively. We have mentioned Ref. [3] (Pohl and Stark, 2014) in our new manuscript and/or SI. We also wish to highlight several key differences between our work and Ref. [3] that may influence dynamic clustering:

- **Polarity of swimmer:** Ref. [3] considers Janus-type phoretic swimmers with inherent polarity, whereas we study isotropic phoretic swimmers that spontaneously develop a swimming orientation via instability. In our case, the polarity of a swimmer can change abruptly due to the influence of surrounding swimmers.
- **Mechanism of attractive and repulsive interactions:** In Ref. [3], the swimmer’s response to the concentration gradient ∇c —via translational/rotational velocity modulates the chemotaxis of swimmers. Translational motion is prescribed to induce attractive interactions, while rotational motion generates either attractive or repulsive interactions. In contrast, our study does not prescribe the kinematics or chemotactic response of our swimmers as they did. Instead, it forms part of the solution, reflecting a Pe-dependent balance between chemical and hydrodynamic interactions. Notably, while our swimmers are chemo-repulsive, their dipolar flow produces hydrodynamic attraction as a signature of pushers (see Our response 9 and Section V of the original SI). At low Pe values, the chemical repulsion leads to the formation of a Wigner crystal. When Pe sufficiently increases, the hydrodynamic attraction overcomes the chemical repulsion, causing the swimmers to dynamically chain together. Finally, at high Pe and area fractions, dynamic clustering emerges.
- **Dry or wet active matter:** Ref. [3] considers point swimmers without considering hydrodynamics. Instead, we account for full hydrodynamic and chemical interactions between finite-sized swimmers.

We did not give the exponent \mathcal{C}_2 of the power-law fitting as the physical implication seems unclear to us. In Fig. 4G of the manuscript, $\mathcal{C}_2 \approx 2$ for Pe = 5 and $\mathcal{C}_2 \approx 1.1$ for Pe = 10.

Despite the value of 2 closely matching the exponent of approximately 2.1 reported in Ref. [3], we choose not to interpret this possible coincidence given the evident discrepancy between the two examined systems.

We now turn our attention to comparing the clustering phenomenon we observe with the experimentally observed clustering of IPAs. We prefer not to label our phoretic disks as “autophoretic”, given that this terminology can also ambiguously refer to Janus swimmers. This term, “autophoretic”, does not appear in our original manuscript or SI either.

As stated in the second paragraph of the original manuscript, and further clarified in Our response 10, IPAs consist of camphor surfers and active droplets. The experiments (Bourgoin et al., 2020) demonstrated the active turbulence formed by camphor surfers, but did not discuss any clustering behavior. In terms of active droplets, we observe three key experiments that have addressed clustering-related phenomena: #1 Thutupalli et al. (2011), #2 Thutupalli et al. (2018), and #3 Hokmabad et al. (2021). The first two experiments showed chain formation in a monolayer of active droplets, tightly confined between two parallel plates. We have compared it with the chain formation we observe in the original SI. The third study investigated active droplets in a deep container, featuring a top air-liquid interface and a bottom flat wall. Clustering was only observed when the droplets either sank to the bottom wall or floated to the top interface due to density mismatch, not when they were density-matched with the solvent. This setup was also examined by experiment #2. In both studies (#2 and #3), the droplets were observed to form layers of crystalline structures, which contrast with the dynamic clustering behavior we report. As suggested by Thutupalli et al. (2018), the fluid motion in the third dimension (normal to the plane) is a significant factor in these observed behaviors, which our two-dimensional simulations cannot capture.

Referee 9: *In the region termed as “active turbulence” here, the authors report formation of irregular chain of particles. How do these chains compare to the ones found earlier in colloidal particles with dipolar interactions, see [4-6].*

[4] Phase diagram of two-dimensional systems of dipole-like colloids
H Schmidle, CK Hall, OD Velev, SHL Klapp - Soft Matter, 2012 - pubs.rsc.org

[5] Collective dynamics of dipolar and multipolar colloids: From passive to active systems
SHL Klapp - Current opinion in colloid & interface science, 2016 - Elsevier

[6] Emergent behavior in active colloids
A Zöttl, H Stark - Journal of Physics: Condensed Matter, 2016 - iopscience.iop.org

Our response 9: We thank the reviewer for mentioning the previously identified chain formation of particles due to dipolar interactions.

Ref. [6] (Zöttl and Stark, 2016) describes the dipolar flow fields of microswimmers but does not mention chain formation: ‘chain’ was not found in that paper, ‘lane formation’ that could be relevant was mentioned once regarding a study on a mixture of passive and active particles. Thus, we will not be able to compare the suggested reference with our results.

Ref. [5] (Klapp, 2016) offers a broad overview of colloids, showcasing numerous dipolar

and multidipolar characteristics associated with various collective behaviours. While this information is comprehensive, a direct comparison between this review compassing many different studies and our work may divert the attention from our core findings. Hence, we aim to distinguish the unique aspects of the dipole-induced chain formation outlined in Ref. [4] (Schmidle et al., 2012) from the chain formation observed in our research, as summarized in Table 1.

Table 1: Comparing the dipole-induced chain formation between Ref. [4] (Schmidle et al., 2012) and our work

	Ref. [4] (Schmidle et al., 2012)	Our work
Source of dipole	Magnetic or electric	Hydrodynamic
Structure of chains	Head-to-tail connection (Fig. 5 of Ref. [4])	Side-by-side connection (Fig. S10 C&D of the SI)
Feature of dipolar interactions	Pairwise; short-ranged	Many-body; long-ranged

We emphasize that the dipolar interaction of our studied phoretic swimmer results from its pusher-type hydrodynamic signature. This has also been identified numerically (Michelin et al., 2013; Morozov and Michelin, 2019), and validated experimentally (Hokmabad et al., 2022). The dipolar flow pattern has been illustrated by the streamlines surrounding a $Pe = 2.5$ swimmer, as shown in the left panel of Fig. 1C of the manuscript. As a pusher, the phoretic disk swimmer draws fluid from the sides, consequently leading to a side-by-side chain forming configuration. This notably contrasts with the head-to-tail chain formation featured in Ref. [4]. Moreover, the low-Reynolds-number hydrodynamic nature of our disk swimmer suggests its the dipolar interaction is characteristically many-bodied and long-ranged. This distinctly contrasts from the dipolar interactions in Ref. [4], which are pairwise and short-ranged.

We have discussed the suggested references in our revised manuscript and/or SI when necessary. The discussion has also been included in the modified SI.

Referee 10: *Comparison to experiments: The authors singled out Refs. [38] and [63] for the comparison of their simulation results with the vast experimental literature on collective behavior of active colloids. Are there other systems in soft matter or in living active systems than camphor boats that are potentially described by the presented model ?*

Our response 10: We would like to underscore that our initial manuscript did not exclusively compare our numerical data with the experimental results in [38] and [63] as the reviewer mentioned. Instead, we did cross-reference several other experiments that share a common feature—the studied phoretic swimmers are isotropic and uniformly emit chemical solutes, as will be proved below.

The second paragraph of our original manuscript introduces that these entities, termed isotropic phoretic agents (IPAs), are exemplified best by active droplets (Thutupalli et al., 2011; Izri et al., 2014; Maass et al., 2016; Moerman et al., 2017; Thutupalli et al., 2018;

Meredith et al., 2020; Lohse and Zhang, 2020; Suda et al., 2021; Dwivedi et al., 2022; Michelin, 2022; Matsuo et al., 2023) and camphor surfers. The model we use (Michelin et al., 2013) has been specifically designed to apply to these two types of IPAs. For the ease of reference, the original, unaltered version is quoted below with certain contents highlighted in bold for better clarity:

“We study a two-dimensional (2D) paradigmatic system of **isotropic phoretic agents (IPAs) represented by active droplets** that have been realized over the last decade [14–24] and **camphor surfers** reported in 1862 [25]. **An IPA, unlike Janus colloids** [26, 27], acquires autonomous propulsion through instability, but otherwise in a stable stationary state. For instance, **an active droplet** undergoing uniform surface reaction exchanges solutes with the ambient, causing an isotropic solute distribution (Fig. 1B). ... **Similarly, this mechanism allows camphor disks to swim** continuously or intermittently [33, 34].”

Furthermore, we did have, in the initial manuscript, compared our results with the experimental observations made on active droplets and camphor surfers. To clarify, we have quoted the relevant sections verbatim below:

- “... creates the active Wigner crystal. This phenomenon agrees with the experimental observations on camphor surfers [38] and **active droplets** [18], as well as numerical prediction for the former [39].”
- “Dynamic arch-shaped chains of closely spaced disks also appear, similar to **chains of active droplets** observed experimentally [14, 18]. ”
- SI, Section IV: “As shown in Fig. S6A (see Movie S9), our simulations reproduce the typical crossing and reflecting trajectories of two phoretic disks previously modelled [19, 20] and **experimentally observed** [20].”
- SI, Section V: “We observed formation of disk chains in gas-like phases when the area fraction $\phi < 0.3$, akin to the experimentally observed counterparts of **chemically active droplets** [23, 24] (Fig. S9). ”

We trust our clarification effectively addresses the reviewer’s concerns. We kindly ask the reviewer to reevaluate their initial recommendation based on our response and the subsequent modifications.

References

- R. Alert, J.-F. Joanny, and J. Casademunt. Universal scaling of active nematic turbulence. Nat. Phys., 16(6):682–688, 2020.
- R. Alert, J. Casademunt, and J.-F. Joanny. Active turbulence. Annu. Rev. Condens. Matter Phys., 13(10.1146), 2022.
- T. Banno, R. Kuroha, and T. Toyota. pH-sensitive self-propelled motion of oil droplets in the presence of cationic surfactants containing hydrolyzable ester linkages. Langmuir, 28(2):1190–1195, 2012.
- J. Bialké, T. Speck, and H. Löwen. Crystallization in a dense suspension of self-propelled particles. Phys. Rev. Lett., 108(16):168301, 2012.
- D. Boniface, C. Cottin-Bizonne, R. Kervil, C. Ybert, and F. Detcheverry. Self-propulsion of symmetric chemically active particles: Point-source model and experiments on camphor disks. Phys. Rev. E, 99(6):062605, 2019.
- M. Bourgoïn, R. Kervil, C. Cottin-Bizonne, F. Raynal, R. Volk, and C. Ybert. Kolmogorovian active turbulence of a sparse assembly of interacting Marangoni surfers. Phys. Rev. X, 10(2):021065, 2020.
- G. Briand and O. Dauchot. Crystallization of self-propelled hard discs. Phys. Rev. Lett., 117(9):098004, 2016.
- C. J. Campbell, E. Baker, M. Fialkowski, and B. A. Grzybowski. Arrays of microlenses of complex shapes prepared by reaction-diffusion in thin films of ionically doped gels. Appl. Phys. Lett., 85(11):1871–1873, 2004.
- M. E. Cates and J. Tailleur. Motility-induced phase separation. Annu. Rev. Condens. Matter Phys., 6:219–244, 2015.
- A. Creppy, O. Praud, X. Druart, P. L. Kohnke, and F. Plouraboué. Turbulence of swarming sperm. Phys. Rev. E, 92(3):032722, 2015.
- P. Dwivedi, D. Pillai, and R. Mangal. Self-propelled swimming droplets. Curr. Opin. Colloid Interface Sci., page 101614, 2022.
- L. Giomi. Geometry and topology of turbulence in active nematics. Phys. Rev. X, 5(3):031003, 2015.
- L. Giomi and M. C. Marchetti. Polar patterns in active fluids. Soft Matter, 8(1):129–139, 2012.
- R. Golestanian. Phoretic active matter. Active matter and nonequilibrium statistical physics, Lecture Notes of the Les Houches Summer School, 112:230–293, 2019.
- C. Gouiller, C. Ybert, C. Cottin-Bizonne, F. Raynal, M. Bourgoïn, and R. Volk. Two-dimensional numerical model of Marangoni surfers: From single swimmer to crystallization. Phys. Rev. E, 104(6):064608, 2021.

- B. I. Halperin and D. R. Nelson. Theory of two-dimensional melting. Phys. Rev. Lett., 41(2):121, 1978.
- J. Happel and H. Brenner. Low Reynolds Number Hydrodynamics: with Special Applications to Particulate Media, volume 1. Springer Science & Business Media, 1983.
- B. V. Hokmabad, R. Dey, M. Jalaal, D. Mohanty, M. Almukambetova, K. A. Baldwin, D. Lohse, and C. C. Maass. Emergence of bimodal motility in active droplets. Phys. Rev. X, 11(1):011043, 2021.
- B. V. Hokmabad, A. Nishide, P. Ramesh, C. Krüger, and C. C. Maass. Spontaneously rotating clusters of active droplets. Soft Matter, 18(14):2731–2741, 2022.
- W.-F. Hu, T.-S. Lin, S. Rafai, and C. Misbah. Chaotic swimming of phoretic particles. Phys. Rev. Lett., 123(23):238004, 2019.
- P. Illien, R. Golestanian, and A. Sen. ‘Fuelled’ motion: phoretic motility and collective behaviour of active colloids. Chem. Soc. Rev., 46(18):5508–5518, 2017.
- Z. Izri, M. N. Van Der Linden, S. Michelin, and O. Dauchot. Self-propulsion of pure water droplets by spontaneous Marangoni-stress-driven motion. Phys. Rev. Lett., 113(24):248302, 2014.
- M. Jiang and Z. Liu. A boundary thickening-based direct forcing immersed boundary method for fully resolved simulation of particle-laden flows. J. Comput. Phys., 390:203–231, 2019.
- M. Jiang, J. Li, and Z. Liu. A simple and efficient parallel immersed boundary-lattice Boltzmann method for fully resolved simulations of incompressible settling suspensions. Comput. Fluids, 237:105322, 2022.
- N. V. Jordan, G. L. Johnson, and A. N. Abell. Tracking the intermediate stages of epithelial-mesenchymal transition in epithelial stem cells and cancer. Cell Cycle, 10(17):2865–2873, 2011.
- B. Kichatov, A. Korshunov, V. Sudakov, V. Gubernov, I. Yakovenko, and A. Kiverin. Crystallization of active emulsion. Langmuir, 37(18):5691–5698, 2021.
- C. Kittel and P. McEuen. Introduction to Solid State Physics. John Wiley & Sons, 2018.
- J. U. Klamser, S. C. Kapfer, and W. Krauth. Thermodynamic phases in two-dimensional active matter. Nat. Commun., 9(1):1–8, 2018.
- S. H. L. Klapp. Collective dynamics of dipolar and multipolar colloids: From passive to active systems. Curr. Opin. Colloid Interface Sci., 21:76–85, 2016.
- J. M. Kosterlitz and D. J. Thouless. Ordering, metastability and phase transitions in two-dimensional systems. Journal of Physics C: Solid State Physics, 6(7):1181, 1973.
- H. Li, S. Li, E. C. Regan, D. Wang, W. Zhao, S. Kahn, K. Yumigeta, M. Blei, T. Taniguchi, K. Watanabe, S. Tongay, A. Zettl, M. F. Crommie, and F. Wang. Imaging two-dimensional generalized Wigner crystals. Nature, 597(7878):650–654, 2021.
- B. Liebchen and A. K. Mukhopadhyay. Interactions in active colloids. J. Phys.: Condens. Matter, 34(8):083002, 2021.

- S.-Z. Lin, W.-Y. Zhang, D. Bi, B. Li, and X.-Q. Feng. Energetics of mesoscale cell turbulence in two-dimensional monolayers. Commun. Phys., 4(1):21, 2021.
- Z. Liu, W. Zeng, X. Ma, and X. Cheng. Density fluctuations and energy spectra of 3d bacterial suspensions. Soft Matter, 17(48):10806–10817, 2021.
- D. Lohse and X. Zhang. Physicochemical hydrodynamics of droplets out of equilibrium. Nat. Rev. Phys., 2(8):426–443, 2020.
- C. C. Maass, C. Krüger, S. Herminghaus, and C. Bahr. Swimming droplets. Annu. Rev. Condens. Matter Phys., 7:171–193, 2016.
- B. Martínez-Prat, R. Alert, F. Meng, J. Ignés-Mullol, J.-F. Joanny, J. Casademunt, R. Golestanian, and F. Sagués. Scaling regimes of active turbulence with external dissipation. Phys. Rev. X, 11(3):031065, 2021.
- M. Matsuo, H. Hashishita, S. Tanaka, and S. Nakata. Sequentially selective coalescence of binary self-propelled droplets upon collective motion. Langmuir, 39(5):2073–2079, 2023.
- C. H. Meredith, P. G. Moerman, J. Groenewold, Y.-J. Chiu, W. K. Kegel, A. van Blaaderen, and L. D. Zarzar. Predator-prey interactions between droplets driven by non-reciprocal oil exchange. Nat. Chem., 12(12):1136–1142, 2020.
- C. H. Meredith, A. C. Castonguay, Y.-J. Chiu, A. M. Brooks, P. G. Moerman, P. Torab, P. K. Wong, A. Sen, D. Velegol, and L. D. Zarzar. Chemical design of self-propelled Janus droplets. Matter, 5(2):616–633, 2022.
- S. Michelin. Self-propulsion of chemically active droplets. Annu. Rev. Fluid Mech., 55, 2022.
- S. Michelin, E. Lauga, and D. Bartolo. Spontaneous autophoretic motion of isotropic particles. Phys. Fluids, 25(6):061701, 2013.
- P. G. Moerman, H. W. Moyses, E. B. Van Der Wee, D. G. Grier, A. Van Blaaderen, W. K. Kegel, J. Groenewold, and J. Brujic. Solute-mediated interactions between active droplets. Phys. Rev. E, 96(3):032607, 2017.
- M. Morozov and S. Michelin. Nonlinear dynamics of a chemically-active drop: From steady to chaotic self-propulsion. J. Chem. Phys., 150(4):044110, 2019.
- S. Nakata, Y. Iguchi, S. Ose, M. Kuboyama, T. Ishii, and K. Yoshikawa. Self-rotation of a camphor scraping on water: new insight into the old problem. Langmuir, 13(16):4454–4458, 1997.
- D. Needleman and Z. Dogic. Active matter at the interface between materials science and cell biology. Nat. Rev. Mater., 2(9):1–14, 2017.
- D. Nishiguchi and M. Sano. Mesoscopic turbulence and local order in Janus particles self-propelling under an ac electric field. Phys. Rev. E, 92(5):052309, 2015.
- J. Palacci, S. Sacanna, A. P. Steinberg, D. J. Pine, and P. M. Chaikin. Living crystals of light-activated colloidal surfers. Science, 339(6122):936–940, 2013.
- A. Pasupalak, Y.-W. Li, R. Ni, and M. P. Ciamarra. Hexatic phase in a model of active biological tissues. Soft Matter, 16(16):3914–3920, 2020.

- Y. Peng, Z. Liu, and X. Cheng. Imaging the emergence of bacterial turbulence: Phase diagram and transition kinetics. Sci. Adv., 7(17):eabd1240, 2021.
- A. P. Petroff, X.-L. Wu, and A. Libchaber. Fast-moving bacteria self-organize into active two-dimensional crystals of rotating cells. Phys. Rev. Lett., 114(15):158102, 2015.
- O. Pohl and H. Stark. Dynamic clustering and chemotactic collapse of self-phoretic active particles. Phys. Rev. Lett., 112(23):238303, 2014.
- K. Qi, E. Westphal, G. Gompper, and R. G. Winkler. Emergence of active turbulence in microswimmer suspensions due to active hydrodynamic stress and volume exclusion. Commun. Phys., 5(1):49, 2022.
- P. J. Schmid and D. S. Henningson. Stability and Transition in Shear Flows. Springer, New York, 2002.
- H. Schmidle, C. K. Hall, O. D. Velev, and S. H. L. Klapp. Phase diagram of two-dimensional systems of dipole-like colloids. Soft Matter, 8(5):1521–1531, 2012.
- R. Singh and R. Adhikari. Universal hydrodynamic mechanisms for crystallization in active colloidal suspensions. Phys. Rev. Lett., 117(22):228002, 2016.
- S. K. Smoukov, K. J. M. Bishop, R. Klajn, C. J. Campbell, and B. A. Grzybowski. Cutting into solids with micropatterned gels. Adv. Mater., 17(11):1361–1365, 2005.
- S. Soh, K. J. M. Bishop, and B. A. Grzybowski. Dynamic self-assembly in ensembles of camphor boats. J. Phys. Chem. B, 112(35):10848–10853, 2008.
- S. Suda, T. Suda, T. Ohmura, and M. Ichikawa. Straight-to-curvilinear motion transition of a swimming droplet caused by the susceptibility to fluctuations. Phys. Rev. Lett., 127(8):088005, 2021.
- T. H. Tan, A. Mietke, J. Li, Y. Chen, H. Higinbotham, P. J. Foster, S. Gokhale, J. Dunkel, and N. Fakhri. Odd dynamics of living chiral crystals. Nature, 607(7918):287–293, 2022.
- S. Thutupalli, R. Seemann, and S. Herminghaus. Swarming behavior of simple model squirmers. New J. Phys., 13(7):073021, 2011.
- S. Thutupalli, D. Geyer, R. Singh, R. Adhikari, and H. A. Stone. Flow-induced phase separation of active particles is controlled by boundary conditions. Proc. Natl. Acad. Sci. USA, 115(21):5403–5408, 2018.
- J. Urzay, A. Doostmohammadi, and J. M. Yeomans. Multi-scale statistics of turbulence motorized by active matter. J. Fluid Mech., 822:762–773, 2017.
- H. H. Wensink, J. Dunkel, S. Heidenreich, K. Drescher, R. E. Goldstein, H. Löwen, and J. M. Yeomans. Meso-scale turbulence in living fluids. Proc. Natl. Acad. Sci. U.S.A., 109(36):14308–14313, 2012.
- A. R. West. Solid State Chemistry and Its Applications. John Wiley & Sons, 2022.
- E. Wigner. On the interaction of electrons in metals. Phys. Rev., 46(11):1002, 1934.

- E. Wigner. Effects of the electron interaction on the energy levels of electrons in metals. Trans. Faraday Soc., 34:678–685, 1938.
- A. P. Young. Melting and the vector Coulomb gas in two dimensions. Phys. Rev. B, 19(4):1855, 1979.
- A. W. Zantop and H. Stark. Emergent collective dynamics of pusher and puller squirmer rods: swarming, clustering, and turbulence. Soft Matter, 18(33):6179–6191, 2022.
- A. Zöttl and H. Stark. Emergent behavior in active colloids. J. Phys. Condens. Matter, 28(25):253001, 2016.
- A. Zöttl and H. Stark. Modeling active colloids: From active Brownian particles to hydrodynamic and chemical fields. Annu. Rev. Condens. Matter Phys., 14:109–127, 2023.

Response to Reviewer II regarding manuscript titled “Shaping active matter: from crystalline solids to active turbulence” authored by Qianhong Yang, Maoqiang Jiang, Francesco Picano, and Lailai Zhu, submitted to Nature Communications

Referee 1: *Qianhong Yang et. al. studied the collective behavior of isotropic phoretic agents. Using large-scale computer simulations, the authors demonstrated solid-liquid-gas phase transitions and also laminar-turbulent transitions in the active system’s fluid phase. This study, I believe, is of significance to the scientific community by providing a paradigmatic framework to advance our understanding of the phase behavior of active suspension, tunable control of active matter and non-equilibrium phenomena in general.*

The authors presentation is succinct and clear. The manuscript is suitable for publication in Nature communications. However, while the premise of this study is of significant importance, I have some comments about the technical details of the manuscript which I would like to have the authors’ response:

Our response 1: We sincerely thank the reviewer for their meticulous evaluation and generous appreciation of our work, as well as for their insightful feedback. In the subsequent sections, we make a concerted effort to address the concerns raised, and implement revisions (marked in magenta) in the manuscript where necessary. We are confident that these adjustments will significantly enhance the clarity and overall quality of our work. Moreover, we have underscored certain parts of the response to facilitate a more streamlined reading experience.

Referee 2: *The individual active particles’ linear translation velocity (U) is central to the analyses of this manuscript. However, the authors do not clearly explained how they calculated the translation (U) and rotational (Ω) velocities neither in the manuscripts’ main text nor in the supporting documentation. The only reference to the translation and rotational velocities relationship with the total force (and torque) acting on the particles is in equations (19a,b) of the Supplemental Information (SI).*

This is perhaps inadequate. U and Ω are unknowns that would only be fixed by imposing the net zero force ($F = 0$) and torque ($L = 0$) constraints. The distinguishing characteristic of active phoretic particles is that the total force (and torque) acting on the phoretic particle (particle + thin/interfacial layer) is zero. That is, $F = 0$ and $L = 0$ irrespective of whether Reynolds number (Re) vanishes or not. $F = 0$ and $L = 0$ are the necessary conditions that would allow us to uniquely determine the particles’ translation and rotational velocities (U and Ω). [see eqns. 3 and 4 in review article ‘<https://doi.org/10.1146/annurev-fluid-120720-012204>’]

Our response 2: We thank the reviewer for pointing this issue, which helps clarify the numerical implementation of our work.

First, we have detailed the calculation of the translational and rotational velocities of disks in the modified SI.

Second, we think that whether using the force/torque-free condition to determine the translational/rotational velocity depends on the Reynolds number of the flow problem. We carefully read the suggested review [1] expressing the force-free condition as Eqs. (3) and (4). This applies there, because the Reynolds number is strictly zero, as shown by the second subequation—steady Stokes equation of Eq. (2). Nevertheless, for finite-Re swimmers such as fishes, Newton’s second law is required to computer their translational/rotational velocity, see e.g., Eq. (13) of Ref. [2] and Eq. (5) of Ref. [3]. The same law also applies for other moving objects such as ships and aeroplanes at a much larger Reynolds number. In such cases, we recognize that the total hydrodynamic/aerodynamic force is equal to the mass of the object multiplied by its instantaneous acceleration rate, which is not zero when the object is accelerating or decelerating. On the other hand, the total force on a zero-Reynolds-number swimmer is zero at any instant regardless of its acceleration rate. Overall, Newton’s second law is the general principle for calculating the translational and rotational velocities of a forced or self-propelling agent. In the limit of Stokes flow, Newton’s second law degenerates to the force- and torque-free conditions, as we will show below.

We consider a three-dimensional object with a characteristic length of a , moving in an incompressible Newtonian fluid. The densities of object and fluid are ρ_o and ρ , respectively. The object’s translational and rotational velocities are $\tilde{\mathbf{U}}$ and $\tilde{\mathbf{\Omega}}$, respectively. The motion of fluid is described by the Navier-Stokes equation for the velocity $\tilde{\mathbf{u}}$ and pressure \tilde{p} ,

$$\tilde{\nabla} \cdot \tilde{\mathbf{u}} = 0, \quad (1a)$$

$$\rho \left(\frac{\partial \tilde{\mathbf{u}}}{\partial \tilde{t}} + \tilde{\mathbf{u}} \cdot \tilde{\nabla} \tilde{\mathbf{u}} \right) = -\tilde{\nabla} \tilde{p} + \mu \tilde{\nabla}^2 \tilde{\mathbf{u}}, \quad (1b)$$

where μ is the dynamic viscosity of the fluid. Here, \sim indicates dimensional variables.

According to Newton’s second law, the hydrodynamic force $\tilde{\mathbf{F}}$ and torque $\tilde{\mathbf{L}}$ on the object are

$$\tilde{\mathbf{F}} = \tilde{m} \frac{d\tilde{\mathbf{U}}}{d\tilde{t}}, \quad (2a)$$

$$\tilde{\mathbf{L}} = \tilde{I} \frac{d\tilde{\mathbf{\Omega}}}{d\tilde{t}}, \quad (2b)$$

where $\tilde{m} = \rho_o \tilde{V}$ and $\tilde{I} = \rho_o \int_{\tilde{V}} \tilde{r}^2 d\tilde{V}'$ denote the mass and the momentum of inertia of the object, respectively (\tilde{r} is the distance to the axis of rotation).

Choosing \tilde{U} , a , $\mu \tilde{U}/a$ as the characteristic velocity, length, and pressure, respectively, we obtain the dimensionless Navier-Stokes equation,

$$\nabla \cdot \mathbf{u} = 0, \quad (3a)$$

$$\text{Re} \left(\frac{\partial \mathbf{u}}{\partial t} + \mathbf{u} \cdot \nabla \mathbf{u} \right) = -\nabla p + \nabla^2 \mathbf{u}, \quad (3b)$$

where $\text{Re} = \rho \tilde{U} a / \mu$ is the Reynolds number. Accordingly, the dimensionless version of Newton’s second law (2) reads

$$\mathbf{F} = \alpha \beta \text{Re} \frac{d\mathbf{U}}{dt}, \quad (4a)$$

$$\mathbf{L} = \alpha \gamma \text{Re} \frac{d\mathbf{\Omega}}{dt}, \quad (4b)$$

where $\alpha = \rho_o/\rho \sim O(1)$ denotes the density ratio between the object and fluid, $\beta = \tilde{V}/a^3 \sim O(1)$ and $\gamma = a^{-5} \int_{\tilde{V}} \tilde{r}^2 d\tilde{V}' \sim O(1)$. Because $d\mathbf{U}/dt \sim O(1)$ and $d\mathbf{\Omega}/dt \sim O(1)$, the dimensionless force $\mathbf{F} = \mathbf{0}$ and torque $\mathbf{L} = \mathbf{0}$ when $\text{Re} = 0$, corresponding to the force- and torque-free conditions in the Stokes flow.

Returning to our work on two-dimensional isotropic phoretic disks with $\rho_o = \rho$, the characteristic velocity, length, and pressure of the disk correspond to $\tilde{U} = \mathcal{AM}/\mathcal{D}$, disk radius a , and $\mu\tilde{U}/a$, respectively. Additionally, the mass and momentum of inertia per unit length of the disk are $\tilde{m} = \pi\rho_o a^2$ and $\tilde{I} = \tilde{m}a^2/2$. As a result, Eq. (4) simplifies to Eq. (19) in the original SI, with $\alpha = 1$, $\beta = \pi$, and $\gamma = \pi/2$.

Referee 3: *Following #1, we also expect full Eulerian description of the dynamics with the velocity U and Ω included in the flow boundary conditions (see eqns. 7a,b of the main text and 18a,b of the SI). Why are U and Ω missing in the boundary conditions?*

Our response 3: We thank the reviewer for pointing out the error in our description. Both \mathbf{U} and Ω appear in the boundary condition as we have correctly implemented numerically. The correctness is indeed supported by the series of validation cases shown in the SI. However, we did not write the mathematical expressions correctly. We have made the correction in the modified manuscript and detailed the Eulerian description in the modified SI.

Referee 4: *I also find it strange the assertion that the gradient of the solute concentration vanish ($\text{grad } c = 0$) mid-way between two particles due to hexagonal symmetry (see the text below eqn. 1 of the main text). The solute concentration contribution from each individual particle is a long-range $1/r$ term similar to electrostatic potential with a gradient decay ($\sim 1/r^2$). Can the authors justify this assertion?*

Our response 4: We thank the reviewer for noting this potentially confusing argument. The confusion might arise from the phrase ‘hexagonal symmetry’. Below, we clarify this confusion with the help of Figure 1 (slightly adapted from Fig. S5 of the original SI), which illustrates the base state of disks forming a stationary hexagonal lattice and the corresponding distribution of solute distribution.

Figure 1: Base state when solute-emitting disks (black circles) form a hexagonal lattice. The colormap shows the distribution of solute concentration. More details are given in the main text.

At this base state, we focus on the four disks labeled 1, 2, 3 and 4 that are arranged clockwise. A solid line connects the centers of disks 1 and 3, and a dashed line connects those of disks 2 and 4. The two lines orthogonal to each other intersect at the red cross. We define the local basis vectors $\mathbf{e}_{\hat{x}}$ and $\mathbf{e}_{\hat{y}}$ that are parallel to the 1-3 and 2-4 lines, respectively.

Clearly, the distribution of solute concentration \tilde{c} is symmetric about the line 2-4. Hence, the profile of $\tilde{c}(\hat{x})$ on the line 1-3 is symmetric about the intersection (cross), leading to $\partial\tilde{c}/\partial\hat{x}|_{\text{cross}} = 0$. Likewise, we obtain $\partial\tilde{c}/\partial\hat{y}|_{\text{cross}} = 0$ by considering the mirror symmetry about the line 1-3.

Because $\mathbf{e}_{\hat{x}}$ and $\mathbf{e}_{\hat{y}}$ are normal to each other, we reach

$$\begin{aligned}\tilde{\nabla}\tilde{c}|_{\text{cross}} &= \partial\tilde{c}/\partial\hat{x}|_{\text{cross}}\mathbf{e}_{\hat{x}} + \partial\tilde{c}/\partial\hat{y}|_{\text{cross}}\mathbf{e}_{\hat{y}} \\ &= \mathbf{0}.\end{aligned}\tag{5}$$

Referee 5: *Distinct from the instability induced propulsion central to the manuscript discussion, two or more chemically active particles with non-zero mobility (M) would always move with finite translation velocity (U) and zero rotational velocity ($\Omega = 0$) due to the spherical symmetry of the particles. This effect is always present irrespective of the value of the Peclet number (Pe). This seems to be completely missing in the manuscripts' discussion.*

Our response 5: We thank the reviewer for mentioning this important implication. We think that the behavior in the proposed scenario hinges on the number n of particles and the boundedness of the domain, which will be discussed in detail below.

Within an unbounded domain, the area or volume fraction ϕ of isotropic phoretic agents (IPAs) tends towards zero, namely, $\phi \rightarrow 0$, a phenomenon due to a finite n (an interpretation we presume aligns with the reviewer's implication). Hence, a pair of identical IPAs invariably maintain a finite translational velocity and experience zero rotational velocity, a consequence of their disrupted spherical or circular symmetry due to the mutual presence, as anticipated by the reviewer and demonstrated by Ref. [4]. When there are more than two IPAs ($n > 2$), they too will engage in translational motion, however, the disappearance of their rotational motion is not a certainty but rather contingent on their initial configuration. Despite the challenge in deciphering their behavior in a general scenario when $n > 2$, we can make a qualitative prediction for three particles ($n = 3$): their rotational motion ceases when their centers align in a straight line or are equidistant (essentially forming a regular triangle), whereas under other conditions, rotation typically ensues.

In the unbounded case as discussed above, the particles will progressively move away from each other over time due to the inherent chemical repulsion. Ultimately, each particle will reach a degree of separation substantial enough to essentially restore an isolated state. At this juncture, the particle motion will once again depend on the Péclet number, Pe .

Unlike the unbounded case characterized by a zero area/volume fraction ϕ , the periodic domain features a finite ϕ —a configuration we have adopted in our work (see Fig. 1A of the manuscript). In such a setting, our simulations show that multiple disks initiate movement irrespective of Pe . However, given a sufficiently low Pe , indicative of suppressed instability,

these disks will eventually self-organize into a stationary crystalline state.

Following the suggestion of reviewer, we have included this implication in the last section of the revised SI.

Referee 6: *I find the use of the word “droplets”, while ignoring the particles’ internal fluid flow, to describe these active particles throughout this manuscript unjustified. Yes, if the capillary number is small and the viscosity of the inner fluid of the “droplet” is high, the authors can justify ignoring the inner fluid flow. However, in that limit (which I presume the authors imply), calling the particles “droplet” could be misleading. The authors could simply maintain the name “isotropic phoretic agents (IPA)” for the chemically active phoretic particles initially introduced in the main text.*

Our response 6: We appreciate the reviewer’s constructive feedback. The term ‘active droplet’ appears in the main text, Materials and Methods, and SI. We will address the usage in each section individually.

In the main text, ‘active droplet’ is employed four times and is not readily replaceable with ‘IPA’. We will examine each usage in turn:

1. “...isotropic phoretic agents (IPAs) represented by active droplets that have been realized over the last decade...”. We believe retaining ‘active droplet’ is necessary for its initial occurrence.
2. “For instance, an active droplet undergoing uniform surface reaction exchanges solutes with the ambient,...” In this context, we are outlining the propulsion mechanism of a real active droplet. Therefore, replacement with ‘IPA’ may not be appropriate.
3. “This phenomenon aligns with the experimental observations on camphor sulfurs [38] and active droplets [18].” Here, the term ‘active droplets’ references real-world laboratory instances, hence it may be more clear to maintain the original term to avert confusion.
4. “Dynamic arch-shaped chains of closely spaced disks also appear, reminiscent of chains of active droplets observed experimentally [14, 18].” Given similar reasons to the previous case, retaining ‘active droplets’ could be necessary.

In the Materials and Methods section, as well as in SI, we maintain the term ‘active droplet(s)’ only when it refers to previously conducted experimental work. Otherwise, it has been replaced by ‘IPA(s)’.

References

- [1] S. Michelin. “Self-propulsion of chemically active droplets”. In: Annu. Rev. Fluid Mech. 55 (2022).
- [2] M. Bergmann and A. Iollo. “Modeling and simulation of fish-like swimming”. In: J. Comput. Phys. 230.2 (2011), pp. 329–348.
- [3] G. Li et al. “Body dynamics and hydrodynamics of swimming fish larvae: a computational study”. In: J. Exp. Biol. 215.22 (2012), pp. 4015–4033.
- [4] B. Nasouri and R. Golestanian. “Exact phoretic interaction of two chemically active particles”. In: Phys. Rev. Lett. 124.16 (2020), p. 168003.

REVIEWER COMMENTS

Reviewer #1 (Remarks to the Author):

This is a review of the revised version of the "Shaping active matter:..." manuscript by Q. Yang et al. submitted for publication in Nature Communications. They authors have replied in some detail to all points raised in my earlier report from June, 13th, 2023 and revised their original manuscript accordingly. Overall, I found their rebuttal of my earlier concerns about the suitability of their manuscript for publication in "Nature Communications" quite convincing. The manuscript is now clearly improved and brings out the central novelty that was expressed in relatively vague terms in the earlier version of the paper much clearer. In particular, the delineation of their findings from earlier results is much improved now. Overall, the novelty of their work lies indeed in the joint (and novel) inclusion of long-range hydrodynamic and chemical interactions which the authors convincingly argue in their reply to my report. Also the analogy between the crystalline phase in their model and the Wigner crystal is a nice and novel aspect here. The discussion as well as the main text clarify these points more clearly now. The only remaining complaint I have is that the introduction is still mostly the same as before, it would benefit the paper as well as the potential readers if the authors already mention some of the aspects summarized so convincingly in their rebuttal of my report, in particular

(i) experimental application beyond boats

(ii) a more precise description of the results e. g. w. r.t. to the Wigner crystal analogy and the unique combination of interactions in their model for active matter

Altogether, I do recommend the publication of the manuscript if the authors will improve a bit on the introduction and bring out the merits of their work in concise fashion in the introduction.

Reviewer #2 (Remarks to the Author):

Dear Editors, Dear Authors,

Having carefully reviewed the responses to my comments and the corresponding changes in the manuscript, I'm satisfied with the current state of the manuscript. Therefore, I believe the manuscripts' results are noteworthy and maybe of significant interest to the broader scientific community.

Reviewer #3 (Remarks to the Author):

The manuscript presents simulations of the collective dynamics of 2D self-propelled phoretic disks. The effect of increasing activity and area fraction is investigated. The authors report behaviors such as crystals, 2D melting and hexatic phase, and turbulence.

This is a nice contribution to the field of active matter. To my knowledge this is the first large-scale simulation of many active droplets with full hydrodynamics. The characterization of the system is exhaustive. It is an impressive technical accomplishment. However, the contribution impact is reduced by 1) its purely theoretical nature: the field of active matter abounds of theoretical papers with loose relevance to experimentally realizable systems, and 2) scarcity of physical insights: the paper is largely descriptive and is not clear what is the fundamental understanding gained from this research.

I have two major concerns:

1) In the Materials and methods section, the authors state that for they solve Stokes Equation Eq. 6, however the computations are done with Reynolds number $Re=0.5$. I am not convinced that inertia is not important. Lowen, JCP (2020), Goto and Tanaka Nat Comm (2015) show that the collective dynamics is very sensitive to inertia. Reeves et al, Comms. Phys. (2021) also have shown that even small amount of inertia can give rise to turbulent flow in a suspension of active particles (spinners), and turbulence does not appear if Re is strictly zero.

2) Regarding the hexatic phase and the KTHNY transition: I am not sure data with range around a decade is sufficient to claim any power or exponential laws. A box size of 100 seems small, especially given the very long range nature of inertialess HD flows in 2D. Have the authors done convergence tests with respect of the box size? Goto and Tanaka(2015) note that $g(r)$ decays faster (exponentially) at long distances because of the finiteness of the domain size, have the authors checked how $g(r)$ behaves as the box increases, r truncated at different values?

Other comments:

Fig 5: claiming power law $E(q) \sim 1/q$ using less than a decade of wavenumbers is a stretch. To prove inverse cascade, the authors should compute the energy flux.

Why should the flow stirred by the squirming droplets behave like active nematics ? Other active systems have shown variety of power laws (e.g. $-8/3$), although again these should be taken with a grain of salt given the limited range of wavenumbers

A relevant recent work: Shi et al. PRL (2023) 131, 108301 DOI: [10.1103/PhysRevLett.131.108301](https://doi.org/10.1103/PhysRevLett.131.108301)

Response to Reviewer I regarding manuscript titled “Shaping active matter from crystalline solids to active turbulence” (originally titled “Shaping active matter: from crystalline solids to active turbulence”; colon removed based on editorial suggestion) authored by Qianhong Yang, Maoqiang Jiang, Francesco Picano, and Lailai Zhu, submitted to Nature Communications

Referee: *This is a review of the revised version of the "Shaping active matter..." manuscript by Q. Yang et al. submitted for publication in Nature Communications. The authors have replied in some detail to all points raised in my earlier report from June, 13th, 2023 and revised their original manuscript accordingly. Overall, I found their rebuttal of my earlier concerns about the suitability of their manuscript for publication in "Nature Communications" quite convincing. The manuscript is now clearly improved and brings out the central novelty that was expressed in relatively vague terms in the earlier version of the paper much clearer. In particular, the delineation of their findings from earlier results is much improved now. Overall, the novelty of their work lies indeed in the joint (and novel) inclusion of long-range hydrodynamic and chemical interactions which the authors convincingly argue in their reply to my report. Also the analogy between the crystalline phase in their model and the Wigner crystal is a nice and novel aspect here. The discussion as well as the main text clarify these points more clearly now. The only remaining complaint I have is that the introduction is still mostly the same as before, it would benefit the paper as well as the potential readers if the authors already mention some of the aspects summarized so convincingly in their rebuttal of my report, in particular*

(i) experimental application beyond boats

(ii) a more precise description of the results e. g. w. r.t. to the Wigner crystal analogy and the unique combination of interactions in their model for active matter

Altogether, I do recommend the publication of the manuscript if the authors will improve a bit on the introduction and bring out the merits of their work in concise fashion in the introduction.

Our response: We sincerely thank the Reviewer for their careful evaluation and generous reconsideration of our last (second) submission. Following their useful suggestions, we have revised the introduction and content, focusing on the relevance with active droplets and Wigner crystal analogy. The unique combination of hydrodynamic and chemical interactions has been emphasized in the last paragraph of the last submission, which has been also stressed in the modified introduction. For the convenience of the Reviewer, the modified parts are reproduced below.

“... Here, we find that remarkably, tuning the activity of a phoretic medium alone can control its solid- liquid-gas phase transitions and subsequently, laminar- turbulent transitions in fluid phases. Through large- scale, agent-resolved simulations, we investigate suspensions of isotropic phoretic agents (IPAs) epitomized by active droplets [14–26] and camphor surfers [27], explicitly resolving their many-body hydrochemical interactions. Our dual consideration of long-range hydrodynamic and chemical interactions enables not only reproducing characteristic collective behaviours of IPAs observed in the lab, but also reconciling seemingly divergent experimental observations—active crystallization or turbulence. The unified

landscape of phoretic collective dynamics is unattainable by resolving either the hydrodynamic or chemical interaction alone.”

“... Importantly, the lattice constant $\ell \approx 5.5$ considerably larger than the disk diameter implies that this solid shares similarity with the Wigner crystal constituting electrons, as theoretically predicted [38, 39] and directly visualized in experiments very recently [40]. Unlike the long-range Coulomb force causing the electronic Wigner crystallization, the chemorepulsion among phoretic disks [21, 27, 41] creates the active Wigner crystal. This phenomenon agrees with the experimental observations on camphor surfers [42] and active droplets [22], as well as numerical predictions for the former [43]. Such active Wigner crystals are distinct from the hexagonal closed-packed crystallization commonly reported in other active suspensions [44–51], where active units tend to achieve the highest packing fraction akin to the atomic arrangement in graphene layers and some metals.”

Response to Reviewer III regarding manuscript titled “Shaping active matter from crystalline solids to active turbulence” (originally titled “Shaping active matter: from crystalline solids to active turbulence”; colon removed based on editorial suggestion) authored by Qianhong Yang, Maoqiang Jiang, Francesco Picano, and Lailai Zhu, submitted to Nature Communications

Referee 1: *The manuscript presents simulations of the collective dynamics of 2D self-propelled phoretic disks. The effect of increasing activity and area fraction is investigated. The authors report behaviors such as crystals, 2D melting and hexatic phase, and turbulence.*

This is a nice contribution to the field of active matter. To my knowledge this is the first large-scale simulation of many active droplets with full hydrodynamics. The characterization of the system is exhaustive. It is an impressive technical accomplishment.

Our response 1: We express our sincere gratitude to the Reviewer for their thorough evaluation and valuable appreciation of our study. We also deeply value their constructive feedback. In the following sections, we address each of the concerns highlighted, and have made revisions (highlighted in magenta) to the manuscript as necessary. We believe these modifications have substantially enhanced both the clarity and the overall robustness of our work. For the Reviewer’s convenience, we have underscored certain parts of the response to facilitate a more streamlined reading experience.

Referee 2: *However, the contribution impact is reduced by 1) its purely theoretical nature: the field of active matter abounds of theoretical papers with loose relevance to experimentally realizable systems, and 2) scarcity of physical insights: the paper is largely descriptive and is not clear what is the fundamental understanding gained from this research.*

Our response 2: We will respond to these two concerns separately.

1) *On theoretical nature:*

We think that the Reviewer might have missed the multiple close connections between our simulations and experimental observations on active droplets and camphor surfers. They are highlighted below.

Firstly, we have numerically demonstrated the self-assembled hexagonal lattices of phoretic disks in Fig. 1D of the original main article (O-MA). Unlike closely-packed crystallization in many active suspensions mimicking canonical crystalline solids, the lattice formed here features an evident inter-atom distance, resembling the Wigner crystal theoretically predicted for electrons and directly visualized recently. Notably, such a Wigner crystalline structure has been clearly identified in camphor surfers (Soh et al., 2008). It is also potentially observed, albeit with some ambiguity due to visualization challenges, in active droplets (Thutupalli et al., 2018). For a clearer comparison, we present in Figure 1 the crystalline structures from these two experiments side by side with our Wigner crystal.

It is noteworthy from Figure 1C that the active droplets appear to be separated by a

Figure 1: A, Wigner crystal of isotropic phoretic agents we observe (taken from Fig. 1D of the O-MA), with an area fraction of $\phi = 0.12$. B, Wigner crystal-like lattice formed by camphor surfers at an air-liquid interface (Soh et al., 2008), where the surfers are clearly separated. C, possible appearance of Wigner crystal assembled by active droplets beneath an air-liquid interface (Thutupalli et al., 2018).

discernible distance. However, it cannot be ruled out that they might have formed a closely-packed crystalline structure. If their interfaces were not visually distinct, it could give the illusion of Wigner-styled separation.

We wish to emphasize that the parallels between our simulations and prior experiments have been discussed in the O-MA, as quoted verbatim below:

“Importantly, the lattice constant $\ell \approx 5.5$ considerably larger than the disk diameter implies that this solid shares similarity with the Wigner crystal constituting electrons, as theoretically predicted [38, 39] and directly visualized in experiments very recently [40]. Unlike the long-range Coulomb force causing the electronic Wigner crystallization, the chemorepulsion among phoretic disks [20, 26, 41] creates the active Wigner. This phenomenon agrees with the experimental observations on camphor surfers [42] and active droplets [21], as well as numerical prediction for the former [43]. It differs from the closely-packed crystallization in other active suspensions [44–50] that mimic canonical crystalline solids.

Secondly, we have identified the crossing and reflecting trajectories of two active droplets, as recently observed experimentally (Hokmabad et al., 2022). The two typical trajectories were shown in Fig. S7 of the original Supplemental Information (O-SI). For clarity, we show in Figure 2 our trajectories with their experimental analogues (Hokmabad et al., 2022).

Thirdly, in Fig. 1F of the O-MA, we depict arc-shaped chains of phoretic disks. These are closely analogous to those of active droplets identified experimentally (Thutupalli et al., 2011, 2018). Further insights into these analogies are detailed in Sec. VI of the O-SI, where we also discuss the underlying mechanism responsible for such chain formations. To facilitate the Reviewer’s understanding, we provide a comparison of the chain formations from our simulations and those observed in experiments in Figure 3.

Most importantly, our work reconcile two seemingly contradicting experimental ob-

Figure 2: A, crossing and reflecting trajectories of two isotropic phoretic disks observed in our simulations, taken from Fig. S7 A of the O-SI. B, crossing (top panel) and reflecting (bottom panel) trajectories of two active droplets, adapted from Fig. 2 of the recent experimental study (Hokmabad et al., 2022).

Figure 3: A, arc-shaped chains develop in the gas-like phase of phoretic disks at a Péclet number $Pe = 3$ and area fraction $\phi = 0.12$, derived from Fig. 1F of the O-MA. B, an arc-shaped chain composed of 15 phoretic disks, taken from Fig. S11C of the O-SI. C and D, chain formations in active droplets observed in experiments, adapted from Thutupalli et al. (2011) and Thutupalli et al. (2018), respectively.

servations on camphor surfers—active crystallization (Soh et al., 2008) or active turbulence (Bourgoin et al., 2020), as discussed in Sec. X of the O-SI. This reconciliation has significantly strengthened the relationship between our simulation-based study and the broader experimental active matter research. That discussion is quoted verbatim below:

“In the main article, we present a comprehensive simulation study that harmonizes the findings of two separate experiments on camphor surfers, which independently observed states of crystallization (Soh et al., 2008) and active turbulence (Bourgoin et al., 2020). The discrepancy between these observed phenomena, we suggest, is largely due to the different levels of phoretic activity, Pe , presented in the two experiments. We will detail this explanation further.

Both experiments employed camphor surfers of a disk shape, albeit with different dimensions. The disk diameter is 1 mm in Soh et al. (2008), whereas it is 5 mm in Bourgoin et al. (2020). Moreover, the height of disk is quite similar in both studies, specifically 500 and 600 μm respectively. Furthermore, both experiments followed the same protocol (Campbell et al., 2004; Smoukov et al., 2005) to fabricate the disks, which should ensure their comparable levels of chemical activity. Besides the disk itself, another factor that can influence its behavior is the depth of the subsurface fluid, which was set at 5 mm in Soh et al. (2008) and at 10 mm in Bourgoin et al. (2020). However, in both scenarios, these depths are considerably greater than three times the radius of the disk. As per the study (Boniface et al., 2019), when the depth of the fluid exceeds this threshold, its influence on the disk’s behavior becomes negligible. Collectively recalling the linear dependence of Pe on the diameter, we thus deduce that the Pe of the latter experiment is approximately five times that of the former one. This trend indeed qualitatively aligns with our predictions (Fig. 2 of the main article), wherein lower activity leads to crystallization and higher activity induces active turbulence.”

In interim summary, we believe that our work has not only reproduced various characteristic behaviours of active droplets and camphor surfers as observed in experiments but also advanced the mechanistic understanding of these phenomena to some extent. On the other hand, we understand that the reason why the Reviewer might have overlooked the connection between our work and experiments is probably because these linkages were mainly discussed in the O-SI due to page limit.

2) *On scarcity of physical insights:*

We agree with the Reviewer on the importance of physical insights. To emphasize, one of the most salient insights offered in our original manuscript is a concise yet mechanistic scaling that aptly delineates the solid-liquid phase transition, as illustrated in the phase diagram, Fig. 2 of the O-MA. While each numerical data point there necessitates the use of hundreds to thousands of processors running over extended periods ranging from days to weeks, we were able to derive a very simple scaling argument predicting the numerical trend. Namely, Eq. (2) of the O-MA shows that the critical $Pe \propto \phi^{1/2}$, which is well aligned with the numerically identified boundary separating the solid and liquid phases in the regime $\phi \lesssim 0.4$. Further elaboration on the far-field assumptions and the symmetry considerations employed in establishing this scaling relationship was provided in Section III of the O-SI. To emphasize this theoretical scaling, we have added the label ‘Theoretical scaling’ in Fig. 2 of the revised main article.

Referee 3: *I have two major concerns:*

1) In the Materials and methods section, the authors state that for they solve Stokes Equation Eq. 6, however the computations are done with Reynolds number $Re = 0.5$. I am not convinced that inertia is not important. Lowen, JCP (2020), Goto and Tanaka Nat Comm (2015) show that the collective dynamics is very sensitive to inertia. Reeves et al, Comms. Phys. (2021) also have shown that even small amount of inertia can give rise to turbulent flow in a suspension of active particles (spinners), and turbulence does not appear if Re is strictly zero.

Our response 3: We concur with the Reviewer that the effect of small yet finite inertia, $Re = 0.5$, on the collective dynamics we observed was not clearly addressed. In the following, we discuss first the implication of the mentioned references and then the inertia effect in our system based on our new data.

1) *Implications of previous studies:*

We realize that Löwen (2020) reviewed how inertia of active particles affects their collective dynamics. The review did not discuss the inertial effect of fluid because it focused on dry active matter composed of macroscopic particles, *e.g.*, , milli-robots propelled by their internal motors or excited by a vibrating plate. Such milli-robots (Narayan et al., 2007; Kudrolli et al., 2008; Deseigne et al., 2010; Weber et al., 2013; Patterson et al., 2017; Junot et al., 2017; Scholz et al., 2018; Deblais et al., 2018) were typically operated in air, featuring extremely short-ranged interaction purely due to collision. This is in stark contrast to our study with long-ranged hydrodynamics playing a vital role.

Figure 4: Effect of fluid inertia, represented by Re , on the collective dynamics of spinning disks, adapted from Figure 7 of Goto and Tanaka (2015). A, effective diffusion coefficient D_{eff} versus Re ; the vertical dashed line indicates the critical $Re \approx 1.4$, below which D_{eff} mildly depends on Re . B, phase diagram of the collective behaviors depending on Re and area fraction Φ ; the vertical dashed line corresponds to $Re \approx 0.6$, below which the collective patterns do not change over Re . For comparison, the chosen inertia level of $Re = 0.5$ in our study is marked by a vertical solid line in A.

Goto and Tanaka (2015) numerically demonstrated that the collective behavior of a two-

dimensional suspension of spinning disks undergoes significant changes when the fluid inertia spans within $\text{Re} \in (0.05, 15)$, as depicted in Figure 4, adapted from Goto and Tanaka (2015). Their study yields crucial implications for our work, revealing a minor dependency of the collective behavior on Re below a threshold of unity. As illustrated in Figure 4A, the effective diffusion coefficient D_{eff} undergoes a modest decline with increasing Re from 0, preceding a pronounced descent at $\text{Re} \approx 1.4$. Concurrently, the collective pattern remains unchanged for $\text{Re} \in (0, 0.6)$, as seen in Figure 4B. This minor dependency on $\text{Re} < O(1)$ suggests that our choice of $\text{Re} = 0.5$ might stand as a reasonable approximation for Stokes flow.

Diving deeper into the comparative discussion with another study by Reeves et al. (2021), we emphasize the similarities between that study and our work. Both settings involve two nonlinear mechanisms, with a major, driving mechanism, complemented by a minor, auxiliary nonlinearity. In Reeves et al. (2021), an electric field E above a threshold E_c represents the driving force, which induces the Quincke rotation of individual agents of the active fluid via an electro-hydrodynamic instability. This driving nonlinearity is quantified by the ratio, $\gamma = E/E_c$, see Table 1 of Reeves et al. (2021). In our case, the chemical phoretic reaction, as the major driver, instigates the spontaneous movement of single disks through a physicochemical hydrodynamic instability, characterized by Pe . Despite the distinct driving nonlinearities, both works share the same auxiliary nonlinearity—fluid inertia indicated by Re , which has been the focus of the Reviewer.

In particular, Reeves et al. (2021) possess the flexibility to entirely remove inertia ($\text{Re} = 0$), a condition unattainable in our simulations, necessitating the maintenance of a finite albeit small Re . They show in their Fig.3 largely different collective patterns when $\text{Re} = 0, 0.01$, and 0.1 , respectively, corresponding to ordered lanes, disordered lanes, and turbulent-like motion. This significant difference has raised the concerns from the Reviewer, questioning the existence of active turbulence as $\text{Re} \rightarrow 0$ in our setting.

We argue that the strong Re -dependency exhibited in the collective patterns observed in Reeves et al. (2021) may not apply to our study. This argument stems from a discernible discrepancy between the two studies—the relative strength of the driving nonlinear (their electro-hydrodynamic or our physicochemical hydrodynamic) effect compared to the inertia-induced nonlinearity. A scenario where the driving nonlinearity is comparable or even weaker than its inertia counterpart could feasibly intensify the impact of Re variations on collective dynamics. Contrarily, a dominant driving nonlinearity could potentially temper the influence of Re . Reeves et al. (2021) mentioned (end of its third paragraph in page 3) that they used $\gamma = 1.1$ throughout their study unless otherwise specified. This choice of nonlinearity, $\gamma = 1.1$, slightly exceeds $\gamma = 1$ corresponding to the onset of the Quincke instability, which can thus be regarded as weak. In contrast, our active turbulence is identified at $\text{Pe} = 20$. This value is significantly above the critical condition— $\text{Pe} \approx 0.5$, above which a single phoretic disk swims autonomously. Consequently, the driving nonlinearity in our setting significantly supersedes the inertia counterpart, where varying Re at the order of unity may yield minimal influence. This assertion is supported by our new simulations as presented below.

2) *Our new efforts on revealing the inertia effect:*

Before presenting our new numerical data, we underscore our previous investigation on how Re influences the propulsion of a single phoretic disk, as detailed in Sec. D2 of the O-SI (page 6). Therein, Fig. S4B indicates that the speed of a swimmer at $Re = 0.5$ deviates slightly from that of a Stokesian counterpart. However, we admit that this indication for single swimmers does not conclusively demonstrate that an inertia level of $Re = 0.5$ has negligible influence on the collective swimming dynamics.

Figure 5: Influence of inertia, Re , on the Eulerian flow field by considering the disk suspension as a continuum active fluid. In addition to our baseline case $Re = 0.5$ demonstrated in the main article, new simulations for $Re = 0.1, 1,$ and 2 have been conducted. Here, $Pe = 20, \phi = 0.5,$ and $L = 200$. A, instantaneous streamlines and vorticity component $\omega_E \cdot \mathbf{e}_z$ of the flow field. B, size distribution of vortices identified using the Okubo-Weiss parameter.

To gain a more definitive insight, we have expanded our simulations to incorporate both smaller and larger Re values relative to our baseline setting of $Re = 0.5$. Specifically, we examine new cases with $Re = 0.1, Re = 1,$ and $Re = 2$. To focus on the Re -dependence, a domain size of $L = 200$ is adopted for all the four cases. We conduct the comparative analysis, showing the instantaneous continuum flow field and the probability density function (PDF) of vortex sizes in Figure 5, the PDF of the disks' velocity components and the longitudinal velocity differences in Figure 6, as well as the kinetic energy spectrum in Figure 7. As shown by Figure 5 and 6, within the studied range of $Re \in [0.1, 2],$ we do not observe qualitative differences in the continuum flow or the statistics of disks' velocities. On the other hand, the energy spectrum $E(q)$ exhibits mild dependence on Re . Notably, Figure 7B shows that the $\sim q^{-1}$ scaling maintains at $Re = 0.1,$ while appearing in the lower wavenumber regime compared to that scaling for $Re = 0.5.$ It is important to note that the q^{-1} scaling line in Fig.5J of the O-MA was INCORRECT (with a wrong slope) in our previous submissions, as realized when we were examining the inertia effect. Details are provided in Figure 9 of this report.

This part of discussion on the effect of finite Re has been provided in Sec. VIII. A of the updated SI.

Figure 6: Similar to Figure 5, while focusing on the statistics of disks' velocities. A and B, PDF of the disk velocity components U and V , respectively. C, PDF of the longitudinal component ΔU^{\parallel} of the velocity difference between two disks separated by a distance $\mathcal{R} = 5$. D, same as C, but for a larger separation $\mathcal{R} = 20$. No qualitative difference can be detected upon varying Re .

Figure 7: Similar to Figure 5, but comparing the kinetic energy spectrum $E(q)$. A, $E(q)$ for the four Re values. The scaling indicators $\sim q$ and $\sim q^{-1}$ are made for $Re = 0.5$. B, $E(q)$ for $Re = 0.1$ along with its own $\sim q^{-1}$ indicator.

Referee 4: 2) Regarding the hexatic phase and the KTHNY transition: I am not sure data with range around a decade is sufficient to claim any power or exponential laws. A box size of 100 seems small, especially given the very long range nature of inertialess HD flows in 2D. Have the authors done convergence tests with respect of the box size? Goto and Tanaka (2015) note that $g(r)$ decays faster (exponentially) at long distances because of the finiteness of the domain size, have the authors checked how $g(r)$ behaves as the box increases, r truncated at different values?

Our response 4: We acknowledge the influence of the domain size on the melting of active particles.

Before our discussion, we stress the stark difference in the characteristic (unit) length of Goto and Tanaka (2015) and our study: it matches a numerical lattice there, in contrast to the disk radius we use. The disk radius $a = 6.4$ (lattice size) is given in the ‘Methods’ section of Goto and Tanaka (2015), allowing for the conversion of their reported lengths into units of disk radius. This recalibration reveals that their observed algebraic decay of $g_6(r)/g(r)$ transitions to a more rapid decline beyond an approximate length of $400/6.4 \approx 94$ disk radii. Besides, their largest domain size is 320 disk radii, with the corresponding results shown in their Figure 5.

Per the insightful feedback from the Reviewer, we have delved into the dependencies of the translational $g_{q_0}(\mathcal{R})$ and orientational $g_6(\mathcal{R})$ correlation functions on domain size L . This analysis was not previously conducted, because the narrow parameter space, $Pe \in [2.35, 2.4]$ (pinpointed in our last submission) occupied by the hexatic phase, necessitates extensive simulations to navigate the Pe space. Despite the substantial cost, we have checked the domain size effect, with our new assessment extending the domain size from $L = 100$ to $L = 200$, as presented below.

Figure 8: Post-review investigation on the effect of domain size L on the KTHNY melting scenario. Here, $L = 200$ as compared to previously used $L = 100$ in Fig. 3 of the O-MA, with the area fraction $\phi = 0.12$ unchanged. A, translational order correlation function $g_{q_0}(\mathcal{R})$ at $Pe = [2.26, 2.3, 2.35]$. B, similar to A, but for the orientational order correlation function $g_6(\mathcal{R})$.

Based on the expanded data utilizing a domain size of $L = 200$, we delineate in Figure 8 the dependency of the translational order correlation function $g_{q_0}(\mathcal{R})$ and the orientational counterpart $g_6(\mathcal{R})$ on L . The spatial decay of these correlation functions does depend on L within the examined range, as implied by the Reviewer. Nonetheless, the core physical picture remains unchanged. Specifically, the successive solid-to-hexatic and hexatic-to-liquid transitions reported previously persist and adhere to the KTHNY framework. Furthermore, the hexatic phase is identified at $Pe = 2.3$, which is slightly below the previously demarcated regime $Pe \in [2.35, 2.4]$ when $L = 100$, reinforcing the solidity of our initial findings.

Because the fundamental findings of our study have remained consistent, we have not expanded the domain size further for a more definitive insight. This decision is informed by the rapidly growing computational cost, which would render the process prohibitively resource-intensive.

The above discussion on the domain size effect has been incorporated in the updated manuscript.

Referee 5: *Other comments:*

Fig 5: claiming power law $E(q) \sim 1/q$ using less than a decade of wavenumbers is a stretch. To prove inverse cascade, the authors should compute the energy flux.

Our response 5:

Before addressing the raised concern, we will draw attention to a correction from our previous submission concerning Fig. 5J. Specifically, the q^{-1} scaling line was inaccurately represented with an incorrect slope. The correction is illustrated in Figure 9; no data is changed. This oversight was identified during our review of the inertial effects on the energy spectrum $E(q)$, see Figure 7.

Figure 9: An error in Fig. 5J of the O-MA, where the q^{-1} scaling line (highlighted within the ellipse) has an incorrect slope. This error has been rectified in the updated version, reproduced in the right panel. All the data remain unchanged.

We acknowledge the concerns raised and address them comprehensively from three perspectives: 1) computational challenges posed by our numerical implementation in accurately determining the energy flux, 2) unique relationship between the inverse cascade and the energy flux for active turbulence, and 3) resonance of our power-law scaling with that from previous studies.

1) *Computational challenges posed by our numerical implementation in accurately determining the energy flux:*

We start with the energy/power balance in physical space, namely, the rate of the kinetic energy of all disks is totally contributed by the power delivered by the fluid onto them. The dimensionless form can be derived as follows:

$$\begin{aligned}
\pi \text{Re} \sum_{k=1}^N \mathbf{U}_k \cdot \frac{d\mathbf{U}_k}{dt} &= \sum_{k=1}^N \left[\int_{S_k} (\mathbf{U}_k - \mathbf{u} + \mathbf{u}) \cdot \boldsymbol{\sigma} \cdot \mathbf{n}_k dS \right] \\
&= - \sum_{k=1}^N \int_{S_k} \mathbf{u}_{\text{slip},k} \cdot \boldsymbol{\sigma} \cdot \mathbf{n}_k dS - \sum_{k=1}^N \int_{S_k} \boldsymbol{\Omega}_k \times (\mathbf{r} - \mathbf{R}_k) \cdot \boldsymbol{\sigma} \cdot \mathbf{n}_k dS \\
&\quad - \text{Re} \int_{\mathcal{V}_f} \mathbf{u} \cdot \left(\frac{\partial \mathbf{u}}{\partial t} + \mathbf{u} \cdot \nabla \mathbf{u} \right) d\mathcal{V} - \int_{\mathcal{V}_f} \boldsymbol{\sigma} : \nabla \mathbf{u} d\mathcal{V} \\
&= - \sum_{k=1}^N \int_{S_k} (\mathbf{I} - \mathbf{n}_k \mathbf{n}_k) \cdot \nabla c \cdot \boldsymbol{\sigma} \cdot \mathbf{n}_k dS - \sum_{k=1}^N \int_{S_k} \boldsymbol{\Omega}_k \times (\mathbf{r} - \mathbf{R}_k) \cdot \boldsymbol{\sigma} \cdot \mathbf{n}_k dS \\
&\quad - \text{Re} \int_{\mathcal{V}_f} \mathbf{u} \cdot \left(\frac{\partial \mathbf{u}}{\partial t} + \mathbf{u} \cdot \nabla \mathbf{u} \right) d\mathcal{V} - \frac{1}{2} \int_{\mathcal{V}_f} \left[\nabla \mathbf{u} + (\nabla \mathbf{u})^\top \right] : \left[\nabla \mathbf{u} + (\nabla \mathbf{u})^\top \right] d\mathcal{V}.
\end{aligned} \tag{1}$$

Here, \mathbf{U}_k and $\boldsymbol{\Omega}_k = \Omega_k \mathbf{e}_z$ are the translational and rotational velocities of the k -th disk, respectively, S_k denotes its surface, \mathbf{n}_k is the outward normal vector pointing from S_k to the fluid, \mathbf{R}_k indicates the disk center, $\mathbf{u}_{\text{slip},k}$ is the phoretic velocity at the surface S_k . Besides, $\boldsymbol{\sigma} = -p\mathbf{I} + \nabla \mathbf{u} + (\nabla \mathbf{u})^\top$ denotes the total stress tensor of the fluid, and \mathcal{V}_f represents the fluid domain excluding the disks. The derivation of Eq. (1) has used the Gauss theorem, the divergence-free condition $\nabla \cdot \mathbf{u} = 0$, and the equality $\nabla \cdot (\mathbf{u} \cdot \boldsymbol{\sigma}) = \frac{1}{2} \left[\nabla \mathbf{u} + (\nabla \mathbf{u})^\top \right] : \left[\nabla \mathbf{u} + (\nabla \mathbf{u})^\top \right] + \mathbf{u} \cdot (\nabla \cdot \boldsymbol{\sigma})$ for incompressible flows, see the dimensional form given for example in page 29 of Happel and Brenner (1983). Rearranging Eq. (1), we obtain

$$\dot{\mathcal{E}}_d + \dot{\mathcal{E}}_f + \mathcal{D}_f + \mathcal{P}_{\text{slip}} + \mathcal{P}_{\text{rot}} = 0, \tag{2}$$

where $\dot{\mathcal{E}}_d = \pi \text{Re} \sum \frac{1}{2} \frac{d\mathbf{U}_k^2}{dt}$ is the rate of change of the disks' kinetic energy, $\dot{\mathcal{E}}_f = \text{Re} \int_{\mathcal{V}_f} \mathbf{u} \cdot \left(\frac{\partial \mathbf{u}}{\partial t} + \mathbf{u} \cdot \nabla \mathbf{u} \right) d\mathcal{V}$ is the rate of change of the fluid's kinetic energy, $\mathcal{D}_f = \frac{1}{2} \int_{\mathcal{V}_f} \left[\nabla \mathbf{u} + (\nabla \mathbf{u})^\top \right] : \left[\nabla \mathbf{u} + (\nabla \mathbf{u})^\top \right] d\mathcal{V}$ is the viscous dissipation within the fluid, $\mathcal{P}_{\text{slip}} = \sum_{k=1}^N \int_{S_k} (\mathbf{I} - \mathbf{n}_k \mathbf{n}_k) \cdot \nabla c \cdot \boldsymbol{\sigma} \cdot \mathbf{n}_k dS$ represents the swimming power related to the phoretic velocity at the disk surface, and $\mathcal{P}_{\text{rot}} = \sum_{k=1}^N \int_{S_k} \boldsymbol{\Omega}_k \times (\mathbf{r} - \mathbf{R}_k) \cdot \boldsymbol{\sigma} \cdot \mathbf{n}_k dS$ indicates the swimming power due to the disk swimmer' rigid-body motion.

To the best of our knowledge, there seems to be no existing formulation analogous to Eq. (1) and Eq. (2) in the literature, specifically applicable to the power/energy balance in wet active matter consisting of finite-sized microswimmers. With the derived formula, we will describe the encountered difficulties of calculating the involved terms after an indepth exploration.

The core of the challenge lies in the spatial resolution limitations inherent to the numerical method utilized in our study, which falls short of the precision required to faithfully represent Eq. (1). We use a Lattice Boltzmann method (LBM) coupled with an immersed

boundary method (IBM) to solve the flow and capture the moving solid-fluid interfaces, as detailed in SI. LBM typically ensures a second-order spatial accuracy in the velocity field \mathbf{u} , degenerating down to first-order around IBM-represented interfaces—disk surfaces here. This degeneration results from the diffused interfaces (with a finite artificial thickness) against the physically sharp interfaces. This reduction in accuracy impacts both the velocity gradient $\nabla\mathbf{u}$ and $\boldsymbol{\sigma} = -p\mathbf{I} + \nabla\mathbf{u} + (\nabla\mathbf{u})^\top$, critical components of Eq. (1), relegating their accuracy to somewhere between zero and first order. It is noted that typical studies do not need such microscopic field data, but rather their macroscopic feature, *e.g.*, the surface integral of $\boldsymbol{\sigma} \cdot \mathbf{n}$ as the total hydrodynamic force on the immersed object. Such surface integrations commonly exhibit a first-to-second order accuracy despite these issues (Jiang and Liu, 2019).

Besides this low spatial resolution of $\nabla\mathbf{u}$, the complexity is further aggravated by a drawback of IBM based on the direct-forcing approach as we adopt—presence of internal flows inside moving objects (Uhlmann, 2005; Jiang and Liu, 2019). This arises as a side effect of IBM’s pivotal advantage, namely, exemption from differentiating internal and external flows. Because the internal flows do not affect the external flows or the kinematics of moving objects that are of the typical interest, their presence does not raise concerns for most studies. Nonetheless, our intended calculation of \mathcal{E}_f and \mathcal{D}_f involving volume integrations with the fluid domain \mathcal{V}_f requires these internal flows to be precisely excluded, which is challenging owing to the diffused IBM-represented interfaces with a finite thickness.

While we acknowledge the considerable challenges in computing the energy flux within our system, it is noteworthy to mention that such calculations are indeed feasible in other contexts of active matter, such as active nematics. Alert et al. (2020) performed such calculations for inertia-less active nematic turbulence, while Koch and Wilczek (2021) did so with a focus on the inertial effect. Employing a Fourier spectral method, both studies conducted two-dimensional simulations for a periodic domain without moving objects. The spectral method allows for highly accurate computation of the velocity gradient $\nabla\mathbf{u}$, while the absence of moving objects obviates the usage of IBM that will cause the above-identified issues.

2) *Unique relationship between the inverse cascade and the energy flux for active turbulence:*

The Reviewer’s suggestion to validate the presence of an inverse energy cascade through scale-to-scale energy transfer is indeed a staple in the analysis of two-dimensional inertial turbulence (Boffetta and Ecke, 2012). However, emerging research in active turbulence seems to present a different narrative (Urzay et al., 2017; Alert et al., 2020; Carenza et al., 2020b,a), indicating the absence of energy transfer across scales despite the manifestation of an inverse power-law scaling.

By assessing the spectral energy budget, these studies deconstruct the energy balance into contributions from various sources: inertial advection, activity, viscous fluid motion, nematic elasticity, and surface friction among others. A pivotal finding from these analyses is that the energy injected into the system at a certain scale is concurrently dissipated at the same scale. This implies there is no residual energy to transition across scales, thus precluding the energy cascade (Alert et al., 2022).

Given this contemporary understanding, it is plausible that the active turbulence observed in our system adheres to a similar framework. Therefore, the energy flux cross scales may not confirm or contradict the reported inverse power-law scaling $E(q) \sim q^{-1}$. We recognize the potential of such a spectral energy analysis (as suggested by the Reviewer) to enhance the physical understanding of active phoretic turbulence. This recognition serves as a strong impetus for us to address and surmount the computational challenges previously outlined.

3) *Resonance of our power-law scaling with that from previous studies:*

Figure 10: Power-law scaling range of energy spectra as presented in well-recognized experimental (A-F) and numerical (G-J) studies on active turbulence. A, suspension of *B. subtilis*, adapted from Wensink et al. (2012). B, suspension of ram spermatozoa, adapted from Creppy et al. (2015). C, electrically driven Janus colloids in a monolayer, adapted from Nishiguchi and Sano (2015). D, three-dimensional suspension of *E. coli*, adapted from Liu et al. (2021). E, monolayers of different tissue cells, adapted from Lin et al. (2021). F, a thin film of microtubules, adapted from Martínez-Prat et al. (2021). G, two-dimensional active nematodynamics, adapted from Giomi (2015). H, three-dimensional active nematodynamics, adapted from Urzay et al. (2017). I, a monolayer suspension of prolate squirmers confined between two plates, adapted from Qi et al. (2022). J, a monolayer of squirming rods within two plates, adapted from Zantop and Stark (2022).

We first underscore that the Reviewer’s impression of the q^{-1} scaling within a decade

roots from the erroneous line indicator (wrong slope) for that scaling drawn in the original Fig.5J. This error has been fixed in the updated manuscript, with the modified Fig.5J given in the right-hand side of Figure 9 of this report. As shown clearly, the data remain unchanged.

After correcting the slope of the q^{-1} scaling line, we have determined that the corrected scaling extends over the wavenumber range of q/q_c from 0.3 to 0.4 orders of magnitude. This extension resonates with the ranges observed in the energy spectra $E(q)$ of previous studies on active turbulence. These prior investigations have been meticulously examined and their results are compiled in Figure 10, providing a comprehensive overview. This summary includes well-recognized experimental (Wensink et al., 2012; Creppy et al., 2015; Nishiguchi and Sano, 2015; Liu et al., 2021; Lin et al., 2021; Martínez-Prat et al., 2021) and numerical (Giomi, 2015; Urzay et al., 2017; Qi et al., 2022; Zantop and Stark, 2022) studies on active turbulence. We draw attention to the above-mentioned six experimental datasets, selected as representative examples by the recent review titled “Active Turbulence”. This review, by Alert et al. (2022), was published in the prestigious journal “Annual Review of Condensed Matter Physics”.

Upon careful scrutiny of the power-law scaling in these ten collected studies, we find that the widest span identified here possibly corresponds to the scaling $E(k) \sim k^{5/3}$, apparent in the orange and red lines in Figure. 10H around $k/k_c = 10^{-1}$ (Qi et al., 2022), and the scaling $E(k) \sim k^{1.4}$, represented by the blue line in Figure. 10I around $k/k_c = 10^{-1}$ (Zantop and Stark, 2022). Both scalings cover approximately 0.6 – 0.7 order of magnitude, while the other presented scalings typically cross a range of 0.2 – 0.3 order of magnitude. Hence, the spanned range of 0.3 – 0.4 order of magnitude in our study seems to be aligned with the general trend seen in previously reported active turbulence.

Referee 6: *Why should the flow stirred by the squirming droplets behave like active nematics? Other active systems have shown variety of power laws (e.g. $-8/3$), although again these should be taken with a grain of salt given the limited range of wavenumbers.*

Our response 6: We thank the Reviewer for highlighting this significant point, which has been the subject of thoughtful consideration. Achieving a rigorous understanding of this similarity is indeed challenging and may exceed the ambit of the current study, particularly as a substantial volume of our results have been relegated to the SI. Nonetheless, we offer a phenomenological rationale that could illuminate this matter.

The very recent review (Alert et al., 2022) classifies active turbulence into two categories based on whether the living individuals are polar or nematic. The isotropic phoretic agents (IPAs) in our study is neither strictly polar or nematic, whereas resembling both active units: An IPA propels autonomously, becoming polar via an instability; whereas it remains stationary and apolar in its stable state, resembling a nematic living unit. The dual similarities of IPAs with the two active units imply why the turbulent-like motion of IPA collectives exhibits distinguishing features of both polar and nematic active turbulence: it emerges following an oscillatory pattern (waves here), characteristic of the former (Giomi and Marchetti, 2012; Alert et al., 2022); its energy spectrum displays the universal scaling (Alert et al., 2020) of

the latter to some extent—as commented by the Reviewer.

The above discussion on the observed similarities was provided in the O-SI.

Referee 7: *A relevant recent work: Shi et al. PRL (2023) 131, 108301 DOI: 10.1103/PhysRevLett.131.108301*

Our response 7: We thank the Reviewer for pointing out the relevant study conducted by Shi et al. (2023), which has been aptly referred to in the section ‘Two-Dimensional Melting via A Hexatic Phase’ of our updated main article. The relevant content is quoted verbatim below:

“This intermediate state corresponds to a hexatic phase between the solid and liquid, as described by the celebrated Kosterlitz, Thouless, Halperin, Nelson, and Young (KTHNY) theory [56–58]. This theory, originally built for equilibrium systems, has also been tested upon non-equilibrium counterparts of active agents [49, 59–62].

References

- R. Alert, J.-F. Joanny, and J. Casademunt. Universal scaling of active nematic turbulence. Nat. Phys., 16(6):682–688, 2020.
- R. Alert, J. Casademunt, and J.-F. Joanny. Active turbulence. Annu. Rev. Condens. Matter Phys., 13(10.1146), 2022.
- G. Boffetta and R. E. Ecke. Two-dimensional turbulence. Annu. Rev. Fluid Mech., 44:427–451, 2012.
- D. Boniface, C. Cottin-Bizonne, R. Kervil, C. Ybert, and F. Detcheverry. Self-propulsion of symmetric chemically active particles: Point-source model and experiments on camphor disks. Phys. Rev. E, 99(6):062605, 2019.
- M. Bourgoïn, R. Kervil, C. Cottin-Bizonne, F. Raynal, R. Volk, and C. Ybert. Kolmogorovian active turbulence of a sparse assembly of interacting Marangoni surfers. Phys. Rev. X, 10(2):021065, 2020.
- C. J. Campbell, E. Baker, M. Fialkowski, and B. A. Grzybowski. Arrays of microlenses of complex shapes prepared by reaction-diffusion in thin films of ionically doped gels. Appl. Phys. Lett., 85(11):1871–1873, 2004.
- L. N. Carenza, L. Biferale, and G. Gonnella. Cascade or not cascade? Energy transfer and elastic effects in active nematics. EPL, 132(4):44003, 2020a.
- L. N. Carenza, L. Biferale, and G. Gonnella. Multiscale control of active emulsion dynamics. Phys. Rev. Fluids, 5(1):011302, 2020b.
- A. Creppy, O. Praud, X. Druart, P. L. Kohnke, and F. Plouraboué. Turbulence of swarming sperm. Phys. Rev. E, 92(3):032722, 2015.
- A. Deblais, T. Barois, T. Guerin, P.-H. Delville, R. Vaudaine, J. S. Lintuvuori, J.-F. Boudet, J.-C. Baret, and H. Kellay. Boundaries control collective dynamics of inertial self-propelled robots. Phys. Rev. Lett., 120(18):188002, 2018.
- J. Deseigne, O. Dauchot, and H. Chaté. Collective motion of vibrated polar disks. Phys. Rev. Lett., 105(9):098001, 2010.
- L. Giomi. Geometry and topology of turbulence in active nematics. Phys. Rev. X, 5(3):031003, 2015.
- L. Giomi and M. C. Marchetti. Polar patterns in active fluids. Soft Matter, 8(1):129–139, 2012.
- Y. Goto and H. Tanaka. Purely hydrodynamic ordering of rotating disks at a finite Reynolds number. Nat. Commun., 6(1):5994, 2015.
- J. Happel and H. Brenner. Low Reynolds Number Hydrodynamics: with Special Applications to Particulate Media, volume 1. Springer Science & Business Media, 1983.
- B. V. Hokmabad, J. Agudo-Canalejo, S. Saha, R. Golestanian, and C. C. Maass. Chemotactic self-caging in active emulsions. Proc. Natl. Acad. Sci. USA, 119(24):e2122269119, 2022.

- M. Jiang and Z. Liu. A boundary thickening-based direct forcing immersed boundary method for fully resolved simulation of particle-laden flows. J. Comput. Phys., 390:203–231, 2019.
- G. Junot, G. Briand, R. Ledesma-Alonso, and O. Dauchot. Active versus passive hard disks against a membrane: Mechanical pressure and instability. Phys. Rev. Lett., 119(2):028002, 2017.
- C.-M. Koch and M. Wilczek. Role of advective inertia in active nematic turbulence. Phys. Rev. Lett., 127(26):268005, 2021.
- A. Kudrolli, G. Lumay, D. Volfson, and L. V. Tsimring. Swarming and swirling in self-propelled polar granular rods. Phys. Rev. Lett., 100(5):058001, 2008.
- S.-Z. Lin, W.-Y. Zhang, D. Bi, B. Li, and X.-Q. Feng. Energetics of mesoscale cell turbulence in two-dimensional monolayers. Commun. Phys., 4(1):21, 2021.
- Z. Liu, W. Zeng, X. Ma, and X. Cheng. Density fluctuations and energy spectra of 3D bacterial suspensions. Soft Matter, 17(48):10806–10817, 2021.
- H. Löwen. Inertial effects of self-propelled particles: From active Brownian to active Langevin motion. J. Chem. Phys., 152(4), 2020.
- B. Martínez-Prat, R. Alert, F. Meng, J. Ignés-Mullol, J.-F. Joanny, J. Casademunt, R. Golestanian, and F. Sagués. Scaling regimes of active turbulence with external dissipation. Phys. Rev. X, 11(3):031065, 2021.
- V. Narayan, S. Ramaswamy, and N. Menon. Long-lived giant number fluctuations in a swarming granular nematic. Science, 317(5834):105–108, 2007.
- D. Nishiguchi and M. Sano. Mesoscopic turbulence and local order in Janus particles self-propelling under an ac electric field. Phys. Rev. E, 92(5):052309, 2015.
- G. A. Patterson, P. I. Fierens, F. S. Jimka, P. G. König, A. Garcimartín, I. Zuriguel, L. A. Pugnaloni, and D. R. Parisi. Clogging transition of vibration-driven vehicles passing through constrictions. Phys. Rev. Lett., 119(24):248301, 2017.
- K. Qi, E. Westphal, G. Gompper, and R. G. Winkler. Emergence of active turbulence in microswimmer suspensions due to active hydrodynamic stress and volume exclusion. Commun. Phys., 5(1):49, 2022.
- C. J. Reeves, I. S. Aranson, and P. M. Vlahovska. Emergence of lanes and turbulent-like motion in active spinner fluid. Commun. Phys., 4(1):92, 2021.
- C. Scholz, M. Engel, and T. Pöschel. Rotating robots move collectively and self-organize. Nat. Commun., 9(1):931, 2018.
- X.-Q. Shi, F. Cheng, and H. Chaté. Extreme spontaneous deformations of active crystals. Phys. Rev. Lett., 131(10):108301, 2023.
- S. K. Smoukov, K. J. M. Bishop, R. Klajn, C. J. Campbell, and B. A. Grzybowski. Cutting into solids with micropatterned gels. Adv. Mater., 17(11):1361–1365, 2005.
- S. Soh, K. J. M. Bishop, and B. A. Grzybowski. Dynamic self-assembly in ensembles of camphor boats. J. Phys. Chem. B, 112(35):10848–10853, 2008.

- S. Thutupalli, R. Seemann, and S. Herminghaus. Swarming behavior of simple model squirmers. New J. Phys., 13(7):073021, 2011.
- S. Thutupalli, D. Geyer, R. Singh, R. Adhikari, and H. A. Stone. Flow-induced phase separation of active particles is controlled by boundary conditions. Proc. Natl. Acad. Sci. USA, 115(21):5403–5408, 2018.
- M. Uhlmann. An immersed boundary method with direct forcing for the simulation of particulate flows. J. Comput. Phys., 209(2):448–476, 2005.
- J. Urzay, A. Doostmohammadi, and J. M. Yeomans. Multi-scale statistics of turbulence motorized by active matter. J. Fluid Mech., 822:762–773, 2017.
- C. A. Weber, T. Hanke, J. Deseigne, S. Léonard, O. Dauchot, E. Frey, and H. Chaté. Long-range ordering of vibrated polar disks. Phys. Rev. Lett., 110(20):208001, 2013.
- H. H. Wensink, J. Dunkel, S. Heidenreich, K. Drescher, R. E. Goldstein, H. Löwen, and J. M. Yeomans. Meso-scale turbulence in living fluids. Proc. Natl. Acad. Sci. U.S.A., 109(36):14308–14313, 2012.
- A. W. Zantop and H. Stark. Emergent collective dynamics of pusher and puller squirmer rods: swarming, clustering, and turbulence. Soft Matter, 18(33):6179–6191, 2022.

REVIEWER COMMENTS

Reviewer #1 (Remarks to the Author):

The authors have incorporated a broader discussion on experimental applications of their work as well as a more detailed representation of the relation of their work to Wigner crystals in the revised manuscript. Overall, I found the manuscript again improved and do now recommend its publication in its present form.

Reviewer #3 (Remarks to the Author):

The authors have done a commendable job addressing my questions. I only have a couple of (seemingly minor) points:

I found the discussion in the authors response about the relative importance of the nonlinearity due to the driving and the nonlinearity due to fluid inertia quite useful. I would recommend the authors include a paragraph about this in the main text, along with the references to the inertia-dominated flows and turbulence studied by Goto & Tanaka, Reeves et al., Kokot et al (PNAS, 2017). I think this will help the reader appreciate the novelty of this work.

Regarding the energy flux: It appears the authors have misunderstood my question. The definition of the energy flux can be found in Alexakis, A. & Biferale, L. Cascades and transitions in turbulent flows. Phys. Rep. 767, 1–101 (2018). If the energy spectrum can be calculated, I don't see a reason why the energy flux as defined in this review can not be calculated.

Clarification: Fig 5j in the main text (Figure 7 (in the response)): what happens when q is between 0.3 and 0.6?

Response to Reviewer III regarding manuscript titled “Shaping active matter from crystalline solids to active turbulence” authored by Qianhong Yang, Maoqiang Jiang, Francesco Picano, and Lailai Zhu, submitted to Nature Communications

Referee 1: *The authors have done a commendable job addressing my questions. I only have a couple of (seemingly minor) points:*

Our response 1: We express our sincere gratitude to the Reviewer for their reevaluation and valuable appreciation of our previous response. Below, we address each of the minor issues raised, and have made revisions (highlighted in magenta) to the manuscript as necessary. For the Reviewer’s convenience, we have underscored certain parts of the response to facilitate a more streamlined reading experience.

Referee 2: *I found the discussion in the authors response about the relative importance of the nonlinearity due to the driving and the nonlinearity due to fluid inertia quite useful. I would recommend the authors include a paragraph about this in the main text, along with the references to the inertia-dominated flows and turbulence studied by Goto & Tanaka, Reeves et al., Kokot et al (PNAS, 2017). I think this will help the reader appreciate the novelty of this work.*

Our response 2: Following the Reviewer’s advice, we have discussed in the main text, the similarity and difference between our work and the three referred studies (Goto and Tanaka, 2015; Kokot et al., 2017; Reeves et al., 2021). The relevant part is quoted below:

“We note that our simulations have incorporated a weak yet finite fluid inertia of $Re = 0.5$ to approximate Stokes flow (Materials and Methods). However, inertia is not the primary factor driving our active turbulence, contrary to other configurations [76–78] featuring a dominant inertial effect. Specifically, Ref. [76] demonstrates that a suspension of rotating disks exhibits consistent collective behaviors at varying Re up to ≈ 0.6 , transitioning to chaos at $Re \gtrsim 5$ (see their Figure 7c).

On the other hand, Ref. [78] highlights the high sensitivity of the emerging active turbulence to Re even below 0.1, unlike the weak Re -dependence depicted here (SI). To rationalize the discrepancy, we first emphasize that both settings involve a driving nonlinearity and an auxiliary counterpart. In that study, an electric field of magnitude \tilde{E} above a threshold \tilde{E}_c drives an electrohydrodynamic instability quantified by $\gamma = \tilde{E}/\tilde{E}_c$. In our setting, the driving nonlinearity arising from the phoretic transport causes a hydrochemical instability characterized by Pe . Despite their different driving mechanisms, both studies feature the same auxiliary nonlinearity: inertia. The reason why the strong Re -dependency in Ref. [78] is absent here stems from a distinction in the strength of the driving nonlinearity relative to the inertial one. When the driving nonlinearity is comparable or even weaker than its inertial counterpart, the driver could feasibly intensify the impact of changing Re , whereas a dominant driving nonlinearity may mitigate this impact. Ref. [78] adopts a nonlinearity level of $\gamma = 1.1$, just above the instability threshold: $\gamma = 1$, representing a weak driving nonlinearity. Conversely, we demonstrate active turbulence at $Pe = 20$, a value far above

the threshold, $Pe \approx 0.5$. Hence, our driving nonlinearity substantially exceeds the inertial counterpart, implying that variations in Re around unity may wield minimal influence, as indeed supported by additional simulations (SI).”

Referee 3: *Regarding the energy flux: It appears the authors have misunderstood my question. The definition of the energy flux can be found in Alexakis, A. & Biferale, L. Cascades and transitions in turbulent flows. Phys. Rep. 767, 1-101 (2018). If the energy spectrum can be calculated, i dont see a reason why the energy flux as defined in this review can not be calculated.*

Our response 3: We thank the Reviewer for highlighting this relevant work (Alexakis and Biferale, 2018), hereafter referred to as AL2018. Upon careful inspection, we realize that the AL2018’s definition of energy flux may not be entirely applicable to our study. AL2018 focuses on the classical turbulent flows of Newtonian fluids, hence defining only the energy spectrum $E(k)$ and energy transfer $T(k, t)$ of the fluid flow. In contrast, we do not examine the turbulent nature of the fluid motion, but focuses solely on the statistics of swimmers, an aspect also explored in other experimental (Nishiguchi and Sano, 2015; Bourgoin et al., 2020; Liu et al., 2021) and numerical studies (Qi et al., 2022; Zantop and Stark, 2022). Hence, our investigation diverges from AL2018 in terms of velocity statistics, exploring Lagrangian turbulence of swimmers as opposed to the Eulerian turbulence of fluid motion as studied by AL2018.

Furthermore, it is noteworthy to discuss the Lagrangian and Eulerian aspects of active turbulence. The experimental study on Kolmogorovian active turbulence of camphor disks by Bourgoin et al. (2020) is particularly relevant. The authors conducted an extensive investigation of Lagrangian turbulence of swimmers, akin to our approach, and also investigated the motion of the surrounding fluid. They concluded that the fluid motion, while not turbulent, can be best described as chaotic (see page 7 in Bourgoin et al. (2020)).

Having discussed the difference between AL2018’s Eulerian turbulence and our Lagrangian counterpart that precludes directly using the formulation of AL2018, we then argue that the formulation cannot be used if we relax the constraint by switching our focus onto the energy transfer within the fluid, as explained below.

The energy transfer $T(k, t)$ given in AL2018 is obtained strictly from the incompressible Newtonian Navier-Stokes equation in the Fourier space, see their equations (10) and (12). Nevertheless, this approach requires significant adaptation when considering dynamics beyond the Newtonian flow itself. For example, in viscoelastic turbulent flows, the energy transfer needs to incorporate the governing equation for viscoelastic stress (Casciola and Angelis, 2007; Fathali and Khoei, 2019). Likewise, our system involves an advection-diffusion equation for the concentration field c , whose gradient ∇c on moving disks drives the flow field. Thus, calculating the energy transfer within the flow necessitates the consideration of the full physicochemical hydrodynamics.

In fact, we could further lower the difficulty by disregarding the physicochemical hydrodynamics and considering the collective motion of squirmers (with fixed surface slip velocities)

in a Newtonian fluid. In theory, this would allow the application of AL2018’s methodology to compute the Eulerian energy transfer. However, this poses a significant numerical challenge. To accurately calculate the energy transfer within the fluid around finite-sized swimmers, it is imperative to precisely define the boundary of fluid domain, which evolves in time as part of the solution. This requirement presents a considerable challenge, especially when we use the diffused immersed boundary method to capture the solid-fluid interfaces.

In summary, three major obstacles preclude the direct application of AL2018’s formulation in calculating the energy transfer in our active turbulence: 1) the distinction between Eulerian and Lagrangian turbulence; 2) the additional complexity introduced by physico-chemistry; and 3) the numerical challenge in accurately determining the time-evolving fluid domain. Importantly, the first aspect presents the most significant challenge.

Referee 4: *Fig 5j in the main text (Figure 7 (in the response))*: what happens when q is between 0.3 and 0.6?

Our response 4: We thank the Reviewer for mentioning this point. We indeed observe the peculiar discontinuity in the energy spectrum $E(q)$ when the scaled wavenumber q/q_c falls within the range of approximately 0.3 to 0.6. Substantial efforts have been subsequently dedicated to identifying the underlying cause of this anomaly, which turn to be rather subtle.

The abnormal $E(q)$ results from one aspect: the canonical expression we have adopted for calculating $E(q)$ is specifically formulated for point microswimmers rather than finite-sized microswimmers as investigated in this study. To elucidate this point, we first recall two classical methods of calculating $E(q)$, both depending on the equal-time two-point velocity correlation function (Eq. 15 of the original SI), $g_{UU}(\mathcal{R})$. Here, \mathcal{R} represents the displacement vector connecting the centers of two swimmers.

The first method consisting in the Fourier transform of $g_{UU}(\mathcal{R})$ reads:

$$E(q) = \left\langle \frac{q}{2\pi} \int \exp(-i\mathbf{q} \cdot \mathcal{R}) g_{UU}(\mathcal{R}) d^2\mathcal{R} \right\rangle_{\theta}, \quad (1)$$

namely, Eq. 16 of the original SI. The second method lies in an angular average of $g_{UU}(\mathcal{R})$ resulting in the one-dimensional velocity correlation function $g_{UU}(\mathcal{R}) = \langle g_{UU}(\mathcal{R}) \rangle_{\theta}$ with $\mathcal{R} = |\mathcal{R}|$. Accordingly, the energy spectrum can be calculated by (Nishiguchi and Sano, 2015):

$$E(q) = q \int_0^{\infty} g_{UU}(\mathcal{R}) \mathcal{R} J_0(q\mathcal{R}) d\mathcal{R}, \quad (2)$$

where J_0 is the zeroth-order Bessel function of the first kind. Notably, the second method assumes an isotropic active turbulence. Our analysis of the data reveals that the two expressions, Eq. (1) and Eq. (2), yield nearly the identical energy spectra $E(q)$ with indiscernible differences.

Evidently, both methods rely on the two-point velocity correlation function, the two-dimensional version $g_{UU}(\mathcal{R})$ in Eq. (1), or, the one-dimensional counterpart $g_{UU}(\mathcal{R})$ in

Eq. (2). In active turbulence of point swimmers, the inter-swimmer distance \mathcal{R} features a lower bound of zero, $\mathcal{R}_{\text{lb}} = 0$. This implies that $g_{UU}(\mathcal{R})$ is physically defined even when two swimmers are infinitely close, $\mathcal{R} \rightarrow 0$. Similarly, in inertial turbulence, the same lower bound of zero permits the computation of velocity correction between any two positions in space, regardless of their proximity. Nevertheless, this scenario is altered for finite-sized swimmers, where the minimum center-to-center distance between swimmers is not zero but a finite value— \mathcal{R}_{lb} —the lower bound. In our setup, this lower bound is $\mathcal{R}_{\text{lb}} = 2$, *viz.*, two radii of phoretic disks. This finite minimum two-point distance in contrast to the zero counterpart in the canonical formula for calculating $E(q)$, Eqs. (1) and (2), causes the abnormal energy spectrum $E(q)$. This anomaly highlights the subtle incompatibility between the classical turbulence theory with the active turbulence in finite-sized active particles. Such an origin of anomaly has not been discussed rigorously in previous studies to the best of our knowledge. After all, active turbulence has only been reported approximately within the last decade.

Even though, among the very few published studies on finite-sized microswimmers' active turbulence, Qi et al. (2022), and Zantop and Stark (2022) reported the “artifact” in their energy spectrum calculations. Both groups overcome this artifact by shifting the correlation function $g_{UU}(\mathcal{R})$ along the \mathcal{R} axis, leading to a shifted correlation function $\hat{g}_{UU}(\hat{\mathcal{R}})$ as a function of a modified inter-particle distance $\hat{\mathcal{R}}$ ¹. Upon such a shift, the lower limit of $\hat{\mathcal{R}} = \mathcal{R} - \mathcal{R}_{\text{lb}}$ becomes zero, which permits using the canonical energy spectrum formula. In addition to these numerical investigations, this shift might also be pertinent in experimental studies on bacterial active turbulence. Through private communication with Dr. Kai Qi, the first author of Qi et al. (2022), we learnt that the shift strategy enabled him and co-authors to reproduce the energy spectra shown in Fig. 4 of Wensink et al. (2012) experimentally demonstrating the active turbulence of *B. subtilis*—inherently finite in size.

Despite circumventing the anomaly or artifact in the energy spectrum, this shifting approach introduces a side effect. The new velocity correlation function $\hat{g}_{UU}(\hat{\mathcal{R}})$ effectively measures the correlation of finite-sized swimmers versus their minimum surface-to-surface distance. This definition of distance differs from the classical center-to-center distance between, either two point-like swimmers showing active turbulence, or two spatial positions in inertial turbulence. The difference suggests that the energy spectrum calculated from $\hat{g}_{UU}(\hat{\mathcal{R}})$ features a new, ‘shifted’ wavenumber

$$\hat{q} = \frac{2\pi}{\hat{\mathcal{R}}} = \frac{2\pi}{\mathcal{R} - \mathcal{R}_{\text{lb}}}, \quad (3)$$

in contrast to the original definition $q = 2\pi/\mathcal{R}$.

In our revision, we have adopted the shifting approach and shown the energy spectrum $E(\hat{q})$ versus the shifted wavenumber \hat{q} . The above discussion has been included in the modified SI.

During the course of pinpointing the reason behind the abnormal energy spectrum, we have identified a flaw in our previous calculation of the velocity correlation function $g_{UU}(\mathcal{R})$

¹Zantop and Stark (2022) mentioned that they followed Qi et al. (2022) to circumvent the artifact.

and consequently that of $E(q)$. Here,

$$g_{UU}(\mathcal{R}) = \frac{\sum_{k=1}^N \sum_{k' \neq k} \mathbf{U}_{k'}(t) \cdot \mathbf{U}_k(t) \delta_{\text{dis}}[\mathcal{R} - (\mathbf{R}_{k'} - \mathbf{R}_k)]}{\sum_{k=1}^N \sum_{k' \neq k} \delta_{\text{dis}}[\mathcal{R} - (\mathbf{R}_{k'} - \mathbf{R}_k)]}, \quad (4)$$

which is typically calculated in a discrete fashion. Namely, the whole \mathcal{R} domain is divided into circular rings with a certain width of $\Delta\mathcal{R}$ (in the radial direction). In our previous calculation, the bin size $\Delta\mathcal{R}$ was not chosen sufficiently small. In the revised calculation, we have adopted $\Delta\mathcal{R} = 0.5$, and we find that $g_{UU}(\mathcal{R})$ does not change significantly even the bin size is reduced to $\Delta\mathcal{R} = 0.1$.

Figure 1: A, $E(\hat{q})$ in the revised main article. Black and gray lines denote $E(\hat{q})$ when the domain size $L = 200$ and $L = 400$, respectively. Here, $Pe = 20$ and $Re = 0.5$. B, $E(\hat{q})$ in the revised SI at $Pe = 20$ with different Reynolds number Re . Here, $E(\hat{q})$ is vertically shifted for a better comparison.

Using the modified bin size and applying the shift approach, the energy spectrum $E(\hat{q})$ shown in Fig. 1A becomes different from the former version (Fig. 5J of the original manuscript). Their differences are summarized below:

- We show the energy spectrum $E(\hat{q})$ versus the new wavenumber \hat{q}/q_c in the revision, instead of the original counterpart q/q_c .
- In the low wavenumber regime, the scaling becomes $E \sim \hat{q}^{4/3}$ as opposed to the previous one $E \sim q^1$.
- In the intermediate wavenumber regime, the scaling becomes $E \sim \hat{q}^{-2/3}$ in contrast to the previous one $E \sim q^{-1}$.
- In the large wavenumber regime, we have identified a new scaling $E \sim \hat{q}^{-5/3}$. While in our last submission, we do not report an evident scaling in that regime.

Considering these differences, we have modified the main article and SI accordingly. In addition, Fig. 1B suggests that the trend of energy spectrum $E(\hat{q})$ does not change significantly when the inertial level Re is varied between $[0.1, 2]$. The reason for the weak dependence on Re has been discussed in our last response letter and included in the revised main article

following the Reviewer's suggestion.

Finally, we would like to express our sincere gratitude for the insightful questions posed by the Reviewer. Since receiving the comments on December 1, 2023, we have dedicated nearly two months to unravel and address the nuanced complexities involved.

In this time, we have come to fully appreciate the 'seemingly minor points' initially mentioned by the Reviewer. The efforts expended during this period have deepened our understanding of the energy spectrum and active turbulence in general. We believe that the subtle mismatch we identified between the classical energy spectrum definition and active turbulence in finite-sized active particles will be useful to the community progressively advancing wet active matter research.

References

- A. Alexakis and L. Biferale. Cascades and transitions in turbulent flows. Phys. Rep., 767: 1–101, 2018.
- M. Bourgoin, R. Kervil, C. Cottin-Bizonne, F. Raynal, R. Volk, and C. Ybert. Kolmogorovian active turbulence of a sparse assembly of interacting Marangoni surfers. Phys. Rev. X, 10(2):021065, 2020.
- C. M. Casciola and E. De Angelis. Energy transfer in turbulent polymer solutions. J. Fluid Mech., 581:419–436, 2007.
- M. Fathali and S. Khoei. Spectral energy transfer in a viscoelastic homogeneous isotropic turbulence. Phys. Fluids, 31(9), 2019.
- Y. Goto and H. Tanaka. Purely hydrodynamic ordering of rotating disks at a finite Reynolds number. Nat. Commun., 6(1):5994, 2015.
- G. Kokot, S. Das, R. G. Winkler, G. Gompper, I. S. Aranson, and A. Snezhko. Active turbulence in a gas of self-assembled spinners. Proc. Natl. Acad. Sci. U.S.A., 114(49): 12870–12875, 2017.
- Z. Liu, W. Zeng, X. Ma, and X. Cheng. Density fluctuations and energy spectra of 3D bacterial suspensions. Soft Matter, 17(48):10806–10817, 2021.
- D. Nishiguchi and M. Sano. Mesoscopic turbulence and local order in Janus particles self-propelling under an ac electric field. Phys. Rev. E, 92(5):052309, 2015.
- K. Qi, E. Westphal, G. Gompper, and R. G. Winkler. Emergence of active turbulence in microswimmer suspensions due to active hydrodynamic stress and volume exclusion. Commun. Phys., 5(1):49, 2022.
- C. J. Reeves, I. S. Aranson, and P. M. Vlahovska. Emergence of lanes and turbulent-like motion in active spinner fluid. Commun. Phys., 4(1):92, 2021.
- H. H. Wensink, J. Dunkel, S. Heidenreich, K. Drescher, R. E. Goldstein, H. Löwen, and J. M. Yeomans. Meso-scale turbulence in living fluids. Proc. Natl. Acad. Sci. U.S.A., 109(36): 14308–14313, 2012.
- A. W. Zantop and H. Stark. Emergent collective dynamics of pusher and puller squirmer rods: swarming, clustering, and turbulence. Soft Matter, 18(33):6179–6191, 2022.

REVIEWERS' COMMENTS

Reviewer #3 (Remarks to the Author):

I am satisfied with the revisions and the authors response.